# Attentional-Biased Stochastic Gradient Descent

**Qi Qi**[1], **Yi Xu**[2], **Wotao Yin**[2], **Rong Jin**[3], **Tianbao Yang**[4]*

*qi-qi@uiowa.edu, statxy@gmail.com, wotao.yin@alibaba-inc.com, rongjinrmail@gamil.com, tianbao-yang@tamu.edu.cn*
[1] *Department of Computer Science, The University of Iowa*
[2] *Alibaba Group*
[3] *Meta Inc*
[4] *Department of Computer Science and Engineering, Texas A&M University*
*\*corresponding author*

**Reviewed on OpenReview:** *https: // openreview. net/ forum? id=B0WYWvVA2r*

## Abstract

In this paper, we present a simple yet effective provable method (named ABSGD) for addressing the data imbalance or label noise problem in deep learning. Our method is a simple modification to momentum SGD where we assign an individual importance weight to each sample in the mini-batch. The individual-level weight of sampled data is systematically proportional to the exponential of a scaled loss value of the data, where the scaling factor is interpreted as the regularization parameter in the framework of distributionally robust optimization (DRO). Depending on whether the scaling factor is positive or negative, ABSGD is guaranteed to converge to a stationary point of an information-regularized min-max or min-min DRO problem, respectively. Compared with existing class-level weighting schemes, our method can capture the diversity between individual examples within each class. Compared with existing individual-level weighting methods using meta-learning that require three backward propagations for computing mini-batch stochastic gradients, our method is more efficient with only one backward propagation at each iteration as in standard deep learning methods. ABSGD is flexible enough to combine with other robust losses without any additional cost. Our empirical studies on several benchmark datasets demonstrate the effectiveness of the proposed method.

## 1 Introduction

Deep Learning (DL) has emerged as the most popular machine learning technique in recent years. It has brought transformative impact in industries and quantum leaps in the quality of a wide range of everyday technologies including face recognition (Schroff et al., 2015; Taigman et al., 2014; Parkhi et al., 2015; Wen et al., 2016; Liu et al., 2019; Qi & Ardeshir, 2023), speech recognition (Graves et al., 2013; Chung et al., 2014; Kim, 2014; Graves, 2013; Ravanelli et al., 2018) and machine translation (Cho et al., 2014; Bahdanau et al., 2014; Sutskever et al., 2014; Luong et al., 2015; Vaswani et al., 2018). Most of these systems are built based on learning a deep neural network (DNN) model from a huge amount of data. However, it has been observed that these deep learning systems could fail in some cases caused by undesirable data distribution, such as data imbalance (Johnson & Khoshgoftaar, 2019; Lin et al., 2017; Chan et al., 2019; Fernández et al., 2018; Huang et al., 2019) and noisy labels in the dataset (FAN et al.; Herzig et al., 2013; Kim et al., 2011). To be more specific, for example, Apple's FaceID (a face recognition system) is much less accurate for recognizing a child than an adult (Bud, 2018), and an autonomous driving car might fail at night under the same road condition (Wakabayashi, 2018). The key factors that cause these problems are (i) the training data sets collected from the real-world are usually follows a highly skewed distribution (e.g., the number of facial images of children are much less than that of adults), and/or contain noisily labelled samples due to the inaccurate annotation process (Zhang et al., 2016); (ii) current deep learning systems are not robust enough to overcome negative influence incurred by the real-world imperfect data as most existing

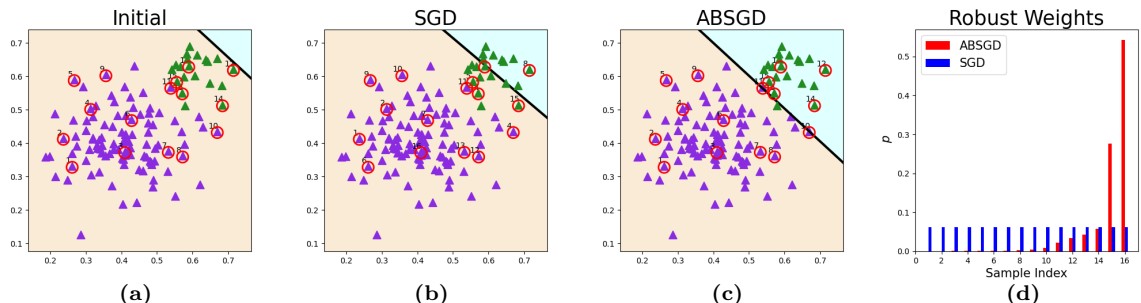

**Figure 1: (a)**: A synthetic data for imbalanced binary classification (green vs purple) with a random linear decision boundary (black line). **(b), (c)**: Learned linear models optimized by SGD and ABSGD with logistic loss for 100 iterations, respectively. **(d)**: The averaged weights of circled samples in the training process of SGD and ABSGD.

deep learning techniques in the literature are crafted and evaluated on well-designed benchmark datasets with balanced distributions among different classes (e.g., ImageNet data for image classification).

Extensive studies have shown that standard empirical risk minimization (ERM) is not sufficient to address the above deficiencies in training on large-scale datasets. Specifically, ERM can lead to biased models toward the majority class and perform poorly in predicting minority class for data imbalance problems (He & Garcia, 2009; Sun et al., 2007; Chawla et al., 2002; Batista et al., 2004). Similarly, label noise can result in incorrect empirical risks during training (Natarajan et al., 2013; Wang et al., 2018; Zhang et al., 2021). To overcome this, recent studies have been explored to learn to improve model robustness to overcome the above deficiencies. Those studies can be divided into two directions, data manipulation, and robust learning. Popular data manipulation methods include under/over-sampling based approaches (Chawla et al., 2002; Han et al., 2005; Chawla, 2009) for data imbalance problem and label correction methods (Xiao et al., 2015; Vahdat, 2017; Lee et al., 2018; Veit et al., 2017) for label noise problem, etc. Existing studies in these methods are not very successful for deep learning with big data. For example, several studies have found that over-sampling yields better performance than using under-sampling (Johnson & Khoshgoftaar, 2019). But over-sampling will add more examples to the training data, which will lead to increased training time. While the label correction methods (Xiao et al., 2015; Vahdat, 2017; Lee et al., 2018; Veit et al., 2017) usually require extra clean data that are expensive to collect.

Robust learning methods include robust weighting and robust loss, where robust weighting assigns weights to different losses of individual data which are either hand-crafted or learned, and robust loss refers to new loss functions that are heuristic-driven or theoretically inspired for addressing data imbalance or label noise issues. The existing robust weighting methods either require significant tuning or suffer from significant computational burden. In this paper, we propose a simple yet systematic **attentional-biased stochastic gradient descent** (ABSGD) method for addressing the class imbalance or the label noise problem in a unified framework, which falls in the category of robust weighting methods. ABSGD is a simple modification of the popular momentum SGD method for deep learning by injecting individual-level importance weights to stochastic gradients in the mini-batch. These importance weights allow our method either to focus on examples from the minority classes for the data imbalance problem or the clean samples for the label noise problem. This idea is illustrated in Figure 1 on a toy imbalanced dataset by comparing it with the standard momentum SGD method for deep learning. Unlike existing meta-learning methods for learning individual-level weights, our individual-level weights are self-adaptive that are computed based on the loss value of each individual data. In particular, the weight for each example is proportional to exponential of a scaled loss value on that example. The weighting scheme is **grounded in the theoretically justifiable distributionally robust optimization (DRO) framework**.

Specifically, our method can be considered as a stochastic momentum method for solving an information-regularized distributionally robust optimization (IR-DRO) problem defined on all possible data (Zhu et al., 2019). From this perspective, our method has several unique features. (i) The weights for all examples in the mini-batch have a proper normalization term to ensure the method optimizes the IR-DRO problem, which is updated online. We prove a theorem to show that our method converges to a stationary solution

of the non-convex IR-DRO problem (with a certain convergence rate). (ii) The scaling factor before the loss value in the exponential function is interpreted as the regularization parameter in the DRO framework. In addition, our method has two benefits: (i) it is applicable in online learning, where the data is received sequentially; (ii) it is loss independent, and can be combined with all existing loss functions crafted for tackling data imbalance and label noise. Finally, we summarize **our contributions** below:

- We propose a simple robust stochastic gradient descent method with momentum and self-adaptive importance weighting to tackle deep learning tasks with imbalanced data or label noise, which is named as **ABSGD**. ABSGD can be generalized to a broader family of AB methods that employ other updating methods, e.g., AB-ADAM that uses the ADAM scheme to update the model parameter.

- We prove that ABSGD finds a stationary solution of a non-convex IR-DRO problem for learning a deep neural network, and establish its convergence rate.

- We compare ABSGD with a variety of existing techniques for addressing the data imbalance and label noise problems, including crafted loss functions, class-balancing weighting methods, individual-level weighting meta-learning methods, and demonstrate superb performance of ABSGD.

## 2   Related Work

**Class-level Weighting**. The idea of class-level weighting is to introduce weights to examples at the class level to balance the contributions from different classes. This idea is rooted in cost-sensitive classification in machine learning (Zhou & Liu, 2005; Sun et al., 2007; Scott, 2011; Parambath et al., 2014; Narasimhan et al., 2015; Elkan, 2001; Busa-Fekete et al., 2015; Yan et al., 2017). Traditional cost-sensitive methods typically tune the class-level weights. Recently, a popular approach is to set the class-wise weights to be proportional to the inverse of class sizes (Huang et al., 2016; Yin et al., 2018). Cui et al. (2019) proposed an improved class-level weighting scheme according to inverse of the "effective number" of examples per class. It is also notable that over/under-sampling methods have the same effect of introducing the class-level weighting to the training algorithm. We can see that these class-level weighting schemes usually require certain knowledge about the size (distribution) of each class, which makes them not suitable to online learning where the size of each class is not known beforehand. These methods also neglect the differences between different examples from the same class (cf. Figure 1).

**Individual-weighting by Meta-Learning**. The individual-level weights learning methods typically use meta-learning to learn the individual-level weights along with updating the model parameters (Jamal et al., 2020; Ren et al., 2018). The idea is to learn individual-level weights by solving a two-level optimization problem. In particular,

$$\min_{\theta} \frac{1}{|\mathcal{C}|} \sum_{\mathbf{z}_i \in \mathcal{C}} L(\mathbf{w}(\theta); \mathbf{z}_i), \quad \text{where } \mathbf{w}(\theta) = \arg\min_{\mathbf{w}} \frac{1}{|\mathcal{D}|} \sum_{\mathbf{z}_i \in \mathcal{D}} \theta_i L(\mathbf{w}; \mathbf{z}_i)$$

where $\mathcal{D}$ denotes the training dataset, $\mathcal{C}$ denotes a balanced validation dataset, $\mathbf{w}$ denotes the model parameter, $\mathbf{z}_i$ denotes a data, $L(\mathbf{w}; \mathbf{z})$ denotes the loss value of model $\mathbf{w}$ on data $\mathbf{z}$, and $\theta = (\theta_1, \ldots, \theta_{|\mathcal{D}|})$ denotes the weights on the training examples. Ren et al. (2018) directly optimized the individual weights in the framework of meta-learning with a heuristic trick by normalizing the weights of all examples in a training batch so that they sum up to one. Jamal et al. (2020) considered the problem from the perspective of domain adaptation and decomposed the individual weight into sum of a non-learnable class-level weight and a learnable individual-level weight. One issue of these meta-learning methods is that they require three back-propagations at each iteration, which is computationally more expensive than our method that is about the same cost of standard SGD for DL.

**Crafted Individual Loss Functions**. Some crafted individual loss functions have been proposed for tackling data imbalance or label noise. A popular loss function is known as the focal loss (Lin et al., 2017), which is a modification of the standard cross-entropy loss. Specifically, it is defined as $-(1 - p_t)^{\gamma} \log(p_t)$ where $\gamma > 0$ is a tuning parameter, $p_t$ is the estimated probability for the ground-truth class. The focal loss has been observed to be effective for dense object detection and is also widely used for classification with

imbalanced data due to its simplicity (Goyal & Kaiming, 2018). However, the focal loss lacks theoretical foundation. To complement this, (Cao et al., 2019) proposed a theoretically-principled label-distribution-aware margin loss, which injects uneven margins into the cross-entropy loss, where the margin for each class is proportional to inverse of each class size to the power of 2/5. For tackling label noise, symmetric losses have been proposed, e.g., symmetric cross entropy loss (SCE) (Wang et al., 2019) and generalized cross entropy loss (TCE) (Zhang & Sabuncu, 2018). Our method is loss independent and hence can be combined with these existing crafted individual loss functions.

**Optimization of DRO**. DRO is a useful technique for domain adaptation, which has been shown both theoretically and empirically promising for learning with imbalanced data (Shalev-Shwartz & Wexler, 2016; Namkoong & Duchi, 2017; Qi et al., 2019; 2022). However, most existing optimization algorithms for DRO are not practical for deep learning, which dims the usefulness of DRO. In the literature, DRO is formulated as (Rahimian & Mehrotra, 2019; Namkoong & Duchi, 2017) :

$$\min_{\mathbf{w} \in \mathbb{R}^d} \max_{\mathbf{p} \in \Delta_n} \sum_{i=1}^{n} p_i L(\mathbf{w}; \mathbf{z}_i) - h(\mathbf{p}, \mathbf{1}/n) + r(\mathbf{w}), \tag{1}$$

where $\Delta_n = \{\mathbf{p} \in \mathbb{R}^n : \sum_i p_i = 1, p_i \geq 0\}$ denotes an $n$-dimensional simplex, $h(\mathbf{p}, \mathbf{1}/n)$ is a divergence measure or constraint between $\mathbf{p}$ and uniform probabilities $\mathbf{1}/n$, $r(\mathbf{w})$ is a standard regularizer on $\mathbf{w}$. We can see DRO aims to minimize the worst-case loss over all the underlying distribution $\mathbf{p}$ in an uncertainty set specified by $h(\mathbf{p}, \mathbf{1}/n)$. Many primal-dual optimization algorithms have been designed for solving the above problem for DL (Rafique et al., 2018; Yan et al., 2020). However, the dual variable $\mathbf{p}$ in the above min-max form is an $n$-dimensional variable restricted to a simplex, which makes existing primal-dual optimization algorithms computationally expensive and not applicable for the online setting where the data is coming sequentially. Our method can be considered as a solution to addressing these issues by considering a specific information-oriented regularizer $h(\mathbf{p}, \mathbf{1}/n) = \lambda \sum_i p_i \log(np_i)$ that is the KL divergence between $\mathbf{p}$ and uniform probabilities $\mathbf{1}/n$, which allows us to transform the min-max formulation into an equivalent minimization formulation with a compositional objective. From this perspective, our method resembles a recently proposed dual-free algorithm RECOVER (Qi et al., 2020). However, RECOVER requires computing stochastic gradients at two different points in each iteration, which causes their GPU cost to double ours. In addition, RECOVER is a variance-reduction method, which might have poor generalization performance. Several recent studies also proposed stochastic algorithms for DRO (Qi et al., 2022; Duchi & Namkoong, 2021; Levy et al., 2020; Jin et al., 2021; Levy et al., 2020; Amid et al., 2019), which are arguably more complicated than our methods.

It was brought to our attention that several papers have developed algorithms based on certain formulations of DRO for tackling noisy data and/or imbalanced data. Li et al. (2021) investigated the effectiveness of optimizing the KL regularized DRO objective in dealing with class imbalance, which is similar to our paper. The difference from this work is that our algorithm is simpler which only uses one mini-batch of samples per-iteration. In contrast, their algorithm requires two independent mini-batches for updating the model parameter. Majidi et al. (2021) proposed an Exponentiated Gradient (EG) reweighting method to optimize the min-min DRO formulation (4) to handle the label noise problem. Unlike that in our algorithm, the normalization term for computing the weights in the mini-batch is simply calculated from the mini-batch, which does not provide any convergence guarantee for solving the min-min DRO formulation. Kumar & Amid (2021) proposed Constrained Instance Weighting (CIW) method that is similar to EG to optimize $f$-divergence min-min DRO, however, no theoretical guarantees have been provided. Later on, Bar et al. (2021) proposed a different algorithm for solving a min-min DRO formulation such that the weights in a constrained simplex $\{\sum_{i=1}^{n} p_i = 1, 0 \leq p_i \leq \mu/n\}$. Their algorithm requires periodic projection onto the constrained simplex, which takes $O(n^2)$ complexity when $\mu < n$ and $O(n)$ complexity when $\mu = n$, where $n$ is the size of the training set. They established a convergence rate of $\sqrt{\frac{n}{B\mathcal{T}}}$, where $B$ denotes the batch size, and $\mathcal{T}$ denote, which is worse than the rate of our proposed algorithm by a factor of $n/B$.

---

**Algorithm 1** ABSGD $(\lambda, \eta, \gamma, \beta, s_0, \mathbf{w}_0, T)$

1: **for** $t = 0, \cdots, T - 1$ **do**
2:   Sample/Receive a mini-batch of $B$ samples $\{\mathbf{z}_1, \cdots, \mathbf{z}_B\}$
3:   Compute $\tilde{g}(\mathbf{w}_t) = \frac{1}{B} \sum_{i=1}^{B} \exp(L(\mathbf{w}_t, \mathbf{z}_i)/\lambda)$
4:   Compute $s_{t+1} = (1 - \gamma)s_t + \gamma \tilde{g}(\mathbf{w}_t)$
5:   Compute $\widetilde{p}_i = \frac{\exp(\frac{L(\mathbf{w}_t; \mathbf{z}_i)}{\lambda})}{s_{t+1}}$, for $i = 1, \ldots, B$
6:   Update $\mathbf{w}_{t+1}$ by Equation (2)
7: **end for**
8: **Return** $\mathbf{w}_T$

---

## 3 Attentional-biased SGD with Momentum (ABSGD)

In this section, we present the proposed method ABSGD and its analysis. We first describe the algorithm and then connect it to the DRO framework. Then we present the convergence result of our method for solving IR-DRO. Throughout this paper, we let $\mathbf{z} = (\mathbf{x}, y)$ denote a random sample that includes an input $\mathbf{x} \in \mathbb{R}^{d'}$ and the class label $y \in \{1, \ldots, K\}$, $\mathbf{w} \in \mathbb{R}^d$ denote the weight of the underlying DNN to be learned. Let $\mathbf{f}(\mathbf{x}) \in \mathbb{R}^K$ be the prediction score of the DNN on data $\mathbf{x}$, and $\ell(\mathbf{f}; y)$ denote a loss function. For simplicity, we let $L(\mathbf{w}; \mathbf{z}) = \ell(\mathbf{f}(\mathbf{x}); y)$. A standard loss function is the cross-entropy loss where $\ell(\mathbf{f}; y) = -\log \frac{\exp(f_y(\mathbf{x}))}{\sum_{k=1}^{K} \exp(f_k(\mathbf{x}))}$. We emphasize that our method is loss independent, and can be applied with any loss functions $\ell(\mathbf{f}; y)$. Specifically, ABSGD can employ the class-level weighted loss functions such as the class-balanced loss (Cui et al., 2019), crafted individual loss functions such as label-distribution aware margin loss (Cao et al., 2019).

### 3.1 Algorithm

The proposed algorithm ABSGD is presented in Algorithm 1. The key steps are described in Step 2 to Step 6, and the key updating step for $\mathbf{w}_{t+1}$ is given by

$$\textbf{ABSGD:} \quad \mathbf{w}_{t+1} = \mathbf{w}_t - \eta \left( \frac{1}{B} \sum_{i=1}^{B} \widetilde{p}_i \nabla L(\mathbf{w}_t; \mathbf{z}_i) + \nabla r(\mathbf{w}_t) \right) + \beta(\mathbf{w}_t - \mathbf{w}_{t-1}) \tag{2}$$

where $r(\mathbf{w}) \propto 1/2 \|\mathbf{w}\|_2^2$ denotes a standard $\ell_2$ norm regularization (i.e., for contributing weight decay in the update). The above update is a simple modification of the standard momentum method (Polyak, 1964), where the last term $\beta(\mathbf{w}_t - \mathbf{w}_{t-1})$ is a momentum term. The modification lies at the introduced weight $\widetilde{p}_i$ for each data $\mathbf{z}_i$ in the mini-batch. The individual weight $\widetilde{p}_i$ is computed in Step 7 and is proportional to $\exp(L(\mathbf{w}_t; \mathbf{z}_i)/\lambda)$, where $\lambda$ is a scaling parameter that $\lambda \in \{\mathbb{R} \backslash 0\}$. Intuitively, we can see that a sample with a large loss value tends to get a higher weight with $\lambda > 0$. It makes sense for learning with imbalanced data since the model tends to fit the data from the majority class while making the loss value larger for the minority class. Hence, the data from the minority class tends to get a larger weight $\widetilde{p}_i$. This phenomenon is demonstrated on a toy dataset in Figure 1. Similarly, if $\lambda < 0$, large value losses have smaller weights. As the noisy samples incurs larger losses than the clean samples, $\widetilde{p}_i$ would further emphasize more on the clean samples with larger weights, hence $\lambda < 0$ is preferred in the presence of label noise.

It is notable that the weight $\widetilde{p}_i$ is properly normalized by dividing a quantity $s_{t+1}$ that is updated online. In particular, $s_{t+1}$ maintains a moving average of the exponential of the scaled loss value on the sampled data (Step 4). It is notable that the normalization does not make the sum of $\widetilde{p}_i$ in the mini-batch equal to 1. We emphasize that this normalization is essential in twofold: (i) it stabilizes the update without causing a significant numerical issue; (ii) it ensures the algorithm converges to a meaningful solution as presented in the next subsection.

### 3.2 Connection with Min-max or Min-min Robust Optimization

In the next subsection, we will show that ABSGD converges to a stationary solution of two robust optimization problems depending on whether $\lambda$ is positive or negative. In particular, given $n$ training samples

$\{\mathbf{z}_1, \ldots, \mathbf{z}_n\}$ we consider the following min-max and min-min robust optimization:

$$\min_{\mathbf{w} \in \mathcal{R}^d} \underbrace{\max_{\mathbf{p} \in \Delta_n} \sum_{i=1}^{n} p_i L(\mathbf{w}; \mathbf{z}_i) - \tau \sum_{i}^{n} p_i \ln(np_i)}_{F_\tau^{(1)}(\mathbf{w})} + r(\mathbf{w}). \tag{3}$$

$$\min_{\mathbf{w} \in \mathcal{R}^d} \underbrace{\min_{\mathbf{p} \in \Delta_n} \sum_{i=1}^{n} p_i L(\mathbf{w}; \mathbf{z}_i) + \tau \sum_{i}^{n} p_i \ln(np_i)}_{F_\tau^{(2)}(\mathbf{w})} + r(\mathbf{w}) \tag{4}$$

where $\tau > 0$ and $\Delta_n$ is a simplex. In the Appendix, we show that $F_\tau^{(1)}(\mathbf{w}) = \tau \log \frac{1}{n} \sum_i \exp(L(\mathbf{w}; \mathbf{z}_i)/\tau) + r(\mathbf{w})$ and $F_\tau^{(2)}(\mathbf{w}) = -\tau \log \frac{1}{n} \sum_i \exp(-L(\mathbf{w}; \mathbf{z}_i)/\tau) + r(\mathbf{w})$. Similar min-max and min-min formulations have been considered in the literature under the framework of tilting log-likelihood (Choi et al., 2000). Recently, there is some renaissance of solving the min-max and min-min formulation in machine learning. For example, the min-max formulation (3) is also closely related to distributionally robust optimization (Namkoong & Duchi, 2017) with a difference that a regularization is imposed on $\mathbf{p}$ instead of a constraint function. The min-min formulation has been considered in (Majidi et al., 2021) for tackling noisy data. Recently, the titled risk functions $F_\tau^{(1)}(\mathbf{w})$ and $F_\tau^{(2)}(\mathbf{w})$ have been also studied in (Li et al., 2021). We describe the difference between ABSGD and the algorithm in (Li et al., 2021) in detail in section 3.5.

By considering the explicit $\tau \sum_i p_i \log(np_i)$ regularizer in the two DRO formulations, our algorithm is applicable to solving the min-max objective (3) by setting $\lambda = \tau$ and the min-min objective (4) by setting $\lambda = -\tau$. When $\tau = +\infty$, $p_i = 1/n$ according to the close form solution derived in Eqn (5). Then above DRO objectives, Eqn (3) and (4), become the standard empirical risk minimization problem: $\min_{\mathbf{w} \in \mathbb{R}^d} \frac{1}{n} \sum_{i=1}^{n} L(\mathbf{w}; \mathbf{z}_i) + r(\mathbf{w})$. When $\tau = 0$, then $\mathbf{p}$ has only one component equal to 1 that corresponds to the data with largest loss value for Eqn (3) and the data with smallest loss value for Eqn (4). Hence, when $\tau \to 0$, DRO objective (3) becomes the maximal loss minimization: $\min_{\mathbf{w} \in \mathbb{R}^d} \max_i L(\mathbf{w}; \mathbf{z}_i) + r(\mathbf{w})$. And when $\tau \to 0$, DRO objective (4) becomes the minimal loss minimization: $\min_{\mathbf{w} \in \mathbb{R}^d} \min_i L(\mathbf{w}; \mathbf{z}_i) + r(\mathbf{w})$. The above maximal loss minimization has been studied for learning with imbalanced data (Shalev-Shwartz & Wexler, 2016). It was shown theoretically to yield better generalization performance than empirical risk minimization for imbalanced data. However, the maximal loss minimization is sensitive to outliers. Hence, by varying the value of $\tau$ we can enjoy the balanced robustness between the imbalanced data and outliers.

### 3.3 Optimization Analysis

It is nice that the DRO formulation is robust to imbalanced data (Eqn (3)) and noisy data (Eqn (4)). However, the min-max/min formulation of DRO is not friendly to the design of efficient optimization algorithms, especially given the constraint $\mathbf{p} \in \Delta_n$. To this end, we transform the min-max/min formulation (3) and (4) into an equivalent minimization formulation following (Qi et al., 2020). In particular, we first compute the optimal solution of dual variable $\mathbf{p}^*$ for the inner maximization/minimization problem given $\mathbf{w}$. By taking the first-derivation of equation (3) and (4) in terms of $\mathbf{p}$ and setting it to zero, *i.e.* $\nabla_{\mathbf{p}} F(\mathbf{w}, \mathbf{p}) = 0$, we have $\mathbf{p}^*$:

$$p_i^* = \frac{\exp(\frac{L(\mathbf{w}; \mathbf{z}_i)}{\lambda})}{\sum_{i=1}^{n} \exp(\frac{L(\mathbf{w}; \mathbf{z}_i)}{\lambda})}, \quad i = 1, \ldots, n \tag{5}$$

where $\lambda = \tau$ for equation (3) and $\lambda = -\tau$ for equation (4). By substituting $\mathbf{p}^*$ back, we obtain the following equivalent minimization formulation:

$$\min_{\mathbf{w} \in \mathbb{R}^d} F_\lambda(\mathbf{w}) = \lambda \log \left( \frac{1}{n} \sum_{i=1}^{n} \exp \left( \frac{L(\mathbf{w}; \mathbf{z}_i)}{\lambda} \right) \right) + r(\mathbf{w}). \tag{6}$$

In an online learning setting, we can further generalize the above formulation as

$$\min_{\mathbf{w} \in \mathbb{R}^d} F_\lambda(\mathbf{w}) = \lambda \log \left( \mathbb{E}_{\mathbf{z}} \exp(L(\mathbf{w}; \mathbf{z})/\lambda) \right) + r(\mathbf{w}). \tag{7}$$

Given the above minimization formulations, our method ABSGD can be considered as a stochastic algorithm for optimizing (6) in offline learning or optimizing (7) in online learning. Our method is rooted in stochastic optimization for compositional optimization that has been studied in the literature (Wang et al., 2017; Ghadimi et al., 2020; Chen et al., 2020; Qi et al., 2020). Intuitively, we can understand our weighting scheme $\widetilde{\mathbf{p}}$ as following. In offline learning with a big data size where $n$ is huge, it is impossible to calculate the $\mathbf{p}^*$ as in (5) at every iteration due to computation and memory limits. As a result, we need to approximate $\mathbf{p}^*$ in a systematic way.

In our method, we use moving average estimate $s_{t+1}$ to approximate the denominator in $\mathbf{p}^*$, i.e., $\frac{1}{n}\sum_{i=1}^{n}\exp(\frac{L(\mathbf{w};\mathbf{z}_i)}{\lambda})$, and use it to compute a scaled weight of data in the mini-batch by Step 5, i.e.,

$$\widetilde{p}_i = \frac{\exp(\frac{L(\mathbf{w}_t;\mathbf{z}_i)}{\lambda})}{s_{t+1}}, \quad i \in \{1, \cdots B\}. \tag{8}$$

More rigorously, our method ABSGD is a stochastic momentum method for solving a compositional problem in the form $f(g(\mathbf{w}))+r(\mathbf{w})$. To this end, we write the objective in (7) as $f(g(\mathbf{w}))+r(\mathbf{w})$, where $f(g) = \lambda\log(g)$ and $g(\mathbf{w}) = \mathbb{E}_{\mathbf{z}}[\exp(L(\mathbf{w};\mathbf{z})/\lambda)]$. The difficulty of stochastic optimization for the compositional function $f(g(\mathbf{w}))$ lies on computing an approximate gradient at $\mathbf{w}_t$. By the chain rule, its gradient is given by $\nabla f(g(\mathbf{w}_t))\nabla g(\mathbf{w}_t) = \frac{\lambda}{g(\mathbf{w}_t)}\nabla g(\mathbf{w}_t)$. To approximate $\nabla f(g(\mathbf{w}_t)) = \frac{\lambda}{g(\mathbf{w}_t)}$, we use a moving average to estimate $g(\mathbf{w}_t)$ inspired by (Wang et al., 2017), which is updated in Step 4 of Algorithm 1, i.e.,

$$s_{t+1} = (1-\gamma)s_t + \gamma\tilde{g}(\mathbf{w}_t), \quad \text{where} \ \ \tilde{g}(\mathbf{w}_t) = \frac{1}{B}\sum_{i=1}^{B}\exp(\frac{L(\mathbf{w}_t,\mathbf{z}_i)}{\lambda}), \{\mathbf{z}_i\} \text{ are random samples.}$$

And $\nabla g(\mathbf{w}_t)$ can be estimated by mini-batch stochastic gradient, i.e.,

$$\nabla\tilde{g}(\mathbf{w}_t) = \frac{1}{B}\sum_{i=1}^{B}\exp(L(\mathbf{w}_t;\mathbf{z}_i)/\lambda)\frac{\nabla L(\mathbf{w}_t;\mathbf{z}_i)}{\lambda}.$$

Hence, the true gradient $\nabla f(g(\mathbf{w}_t))\nabla g(\mathbf{w}_t)$ is able to be approximated by

$$\frac{\lambda}{s_{t+1}}\nabla\tilde{g}(\mathbf{w}_t) = \frac{1}{B}\sum_{i=1}^{B}\frac{1}{s_{t+1}}\exp(L(\mathbf{w}_t;\mathbf{z}_i)/\lambda)\nabla L(\mathbf{w}_t;\mathbf{z}_i),$$

which is exactly the approximate gradient used in the update of $\mathbf{w}_{t+1}$ as in equation (2). Let us provide an intuition about the benefit of using $s_{t+1}$ for normalization of weights. Let us consider a simple case such that only one data is sampled for updating. For the imbalanced data setting, if the sampled data denoted by $\mathbf{z}_t$ at the $t$-th iteration is from a minority group and hence has a large loss. We would like to penalize more on such an example. The estimator $s_{t+1} = (1-\gamma)s_t + \gamma\exp(L(\mathbf{w}_t,\mathbf{z}_t)/\lambda)$ is likely to be smaller than $\exp(L(\mathbf{w}_t,\mathbf{z}_t)/\lambda)$ due to $\gamma < 1$. As a result, normalization using $s_{t+1}$ will give a larger weight to the sampled minority data compared with using the mini-batch normalization, i.e., $\frac{\exp(L(\mathbf{w}_t,\mathbf{z}_t)/\lambda)}{s_{t+1}} > \frac{\exp(L(\mathbf{w}_t,\mathbf{z}_t)/\lambda)}{\exp(L(\mathbf{w}_t,\mathbf{z}_t)/\lambda)}$. Similarly, in the noisy label setting, if the sampled data is a noisy sample and hence has a large loss, then $\exp(L(\mathbf{w}_t,\mathbf{z}_t)/\lambda)$ would be small due to that $\lambda$ is set to be negative in this case. As a result $s_{t+1} = (1-\gamma)s_t + \gamma\exp(L(\mathbf{w}_t,\mathbf{z}_t)/\lambda)$ is likely to be larger than $\exp(L(\mathbf{w}_t,\mathbf{z}_t)/\lambda)$. Then normalization using $s_{t+1}$ will give a smaller weight to the sampled noisy data compared with using the mini-batch normalization, i.e., $\frac{\exp(L(w_t,x_t)/\lambda)}{s_{t+1}} < \frac{\exp(L(\mathbf{w}_t,\mathbf{z}_t)/\lambda)}{\exp(L(\mathbf{w}_t,\mathbf{z}_t)/\lambda)}$. We can see that in both cases using $s_{t+1}$ for normalization intuitively makes sense. In Figure 4, we will empirically demonstrate the benefit of our algorithm design with a variant that uses $\gamma = 1$ which is just using the mini-batch normalization.

Finally, we comment on the final update of model parameters. Instead of directly using this gradient estimator to update the model parameter following (Wang et al., 2017), we employ a momentum update. The reason is that the algorithm in (Wang et al., 2017) has a larger sample complexity, which is $O(1/\epsilon^5)$ for finding an $\epsilon$-stationary point of the objective function (cf. Section 3.5). By using a momentum update as

in (2), we are able to establish an optimal sample complexity. It is notable that the momentum update can be seen as a simplification of the NASA method proposed in (Ghadimi et al., 2020), which was designed to address the constrained compositional optimization.

### 3.4 Other AB methods

In light of the discussion about the connection between ABSGD and optimization of IR-DRO, we can generalize ABSGD to employ other updating schemes, e.g., AdaGrad (Duchi et al., 2011), RMSProp (Ruder, 2016; Guo et al., 2021), Adam (Kingma & Ba, 2015). Below, we present the ABAdam method. The key idea is to replace the standard mini-batch gradient estimator in Adam by the weighted mini-batch gradient estimator. The key steps of ABAdam are presented below.

$$
\begin{aligned}
G(\mathbf{w}_t) &= \frac{1}{B}\sum_{i=1}^{B}\widetilde{p}_i\nabla L(\mathbf{w}_t;\mathbf{z}_i) \\
\textbf{ABAdam:} \quad \mathbf{v}_{t+1} &= \beta_1\mathbf{v}_t + (1-\beta_1)G(\mathbf{w}_t) \\
\mathbf{u}_{t+1} &= \beta_2\mathbf{u}_t + (1-\beta_2)(G(\mathbf{w}_t))^2 \\
\mathbf{w}_{t+1} &= \mathbf{w}_t - \eta\left(\frac{\mathbf{v}_{t+1}}{\sqrt{\mathbf{u}_{t+1}}+G_0} + \nabla r(\mathbf{w}_t)\right)
\end{aligned}
\tag{9}
$$

where $G_0$ is a constant to increase stability, $\beta_1,\beta_2$ are the constant hyperparameters that are usually set as 0.9 and 0.999, respectively. ABAdam could potentially benefit the applications that Adam has better generalization performance than the SGD (Nadkarni et al., 2011; Kang et al., 2020). In the appendix, we provide a theoretical analysis for ABAdam for optimizing IR-DRO, and leave the experimental exploration for the future.

### 3.5 Convergence Analysis

In this subsection, we provide a convergence result of ABSGD for solving the min-max or the min-min objective under some standard assumptions in non-convex optimization. For presentation simplicity, we use the notations $g(\mathbf{w}) = \mathbb{E}_{\mathbf{z}}[\exp(L(\mathbf{w};\mathbf{z})/\lambda)]$ and $g(\mathbf{w};\mathbf{z}) = \exp(L(\mathbf{w};\mathbf{z})/\lambda)$. We first state a standard assumption (Qi et al., 2020; Wang et al., 2017) and then present our main theorem.

**Assumption 1.** *Let $V_g$, $L_l$ are constant scalars,*

- *For a fixed $\lambda$, there exists $V_g > 0$ such that $\mathbb{E}_{\mathbf{z}}[\|g(\mathbf{w};\mathbf{z})-g(\mathbf{w})\|^2] \leq V_g$, $\mathbb{E}_{\mathbf{z}}[\|\nabla g(\mathbf{w};\mathbf{z})-\nabla g(\mathbf{w})\|^2] \leq V_g$ and $L(\mathbf{w};\mathbf{z})$ for any $\mathbf{z}$ is an $L_l$-smooth function, i.e., $\|\nabla L(\mathbf{w};\mathbf{z}) - \nabla L(\mathbf{w}';\mathbf{z})\| \leq L_l\|\mathbf{w}-\mathbf{w}'\|, \forall \mathbf{w}, \mathbf{w}'$*

- *For a given $\tau$, there exists $\Delta_0$ such that $F_\tau^{(1)}(\mathbf{w}_1) - \min F_\tau^{(1)}(\mathbf{w}) \leq \Delta_0$ or $F_\tau^{(2)}(\mathbf{w}_1) - \min F_\tau^{(2)}(\mathbf{w}) \leq \Delta_F$.*

**Theorem 1.** *Assume assumption 1 holds and there exists $C_0, C_1$ such that $\exp(L(\mathbf{w}_t;\mathbf{z}_i)/\lambda) \leq C_0, \|\nabla L(\mathbf{w}_t;\mathbf{z}_i)\| \leq C_1$, for all $\mathbf{w}_t$ and any $\mathbf{z}_i$. Then, $\gamma \leq \frac{\epsilon^2}{3(4G^2+5GV_g)}, \eta = \frac{\gamma^2}{2\sqrt{L_F^2+10GL_g^2}}, \beta = 1-\gamma$. For $\lambda = \tau > 0$, ABSGD ensures that $E\left[\frac{1}{T}\sum_{t=1}^{T}\|\nabla F_\tau^{(1)}(\mathbf{w}_t)\|^2\right] \leq \epsilon^2$ after $T = O(1/\epsilon^4)$ iterations, and for $\lambda = -\tau < 0$, ABSGD ensures that $\mathbb{E}\left[\frac{1}{T}\sum_{t=1}^{T}\|\nabla F_\tau^{(2)}(\mathbf{w}_t)\|^2\right] \leq \epsilon^2$ after $T = O(1/\epsilon^4)$ iterations, where we exhibit the constant in the big O in Appendix.*

**Remark:** Before ending this section, we present some remarks. First, we notice that in a concurrent work (Li et al., 2021), the authors proposed a similar algorithm to ABSGD without the momentum term, i.e., $\gamma = 1$. However, they only prove the convergence for the algorithm with independent mini-batches for $L(\mathbf{w};\mathbf{z})$ and $\nabla L(\mathbf{w};\mathbf{z}')$. In our experiments, we show that the momentum term is important for speeding up the convergence. In another concurrent work (Majidi et al., 2021) the authors proposed an algorithm for solving the min-min objective (4). The difference is that in their algorithm the normalization for computing

the weight is computed only from the current mini-batch while that in ABSGD depends on all historical data. In addition, (Majidi et al., 2021) provides no convergence analysis for solving the min-min robust optimization problem.

### 3.6 Two-stage Training Strategy for $\lambda$

Since $\lambda$ can be interpreted as the regularization parameter in IR-DRO, we can understand its impact on the learning of model. With a larger $|\lambda|$, the IR-DRO is getting closer to ERM and ABSGD is getting close to the standard momentum SGD method without robust weighting. When $|\lambda| = \infty$, the update becomes exactly the same as the standard momentum SGD method. When $|\lambda|$ becomes smaller, the update will focus more on data with larger loss values (e.g., from the minority class). This uneven weighting is helpful to learn a robust classifier. However, it might harm the learning of feature extraction layers. This phenomenon has been also observed in previous works (Cao et al., 2019; Kang et al., 2019).

To address this issue, we employ a two-stage training method following the existing literature (Kang et al., 2019), where in the first stage we employ momentum SGD to learn a basis network, and in the second stage we employ ABSGD to learn the classifier and finetune the feature layers. As momentum SGD is a special case of ABSGD with $|\lambda| = \infty$, the two-stage method is equivalent to running ABSGD with $|\lambda| = \infty$ first and then restarting it with a decayed $|\lambda| < \infty$. In the ablation study, we will show that damping $|\lambda|$ is critical for balancing the learning of feature extraction layers and classifier layers. Finally, it is notable that in the second stage, we can fix some lower layers and only fine-tune upper layers using ABSGD.

## 4 Experimental Results on Data Imbalance Problem

We conduct experiments on multiple imbalanced benchmark datasets, including CIFAR-10 (LT), CIFAR-10 (ST), CIFAR-100 (LT), CIFAR-100 (ST), ImagetNet-LT (Liu et al., 2019), Places-LT (Zhou et al., 2017), and iNaturelist2018 (iNatrualist, 2018), and compare ABSGD with several state-of-the-art (SOTA) methods, including meta-learning (Jamal et al., 2020), class-balanced weighting (Cao et al., 2019), and two-stage decoupling methods (Kang et al., 2019). We use the ResNets (He et al., 2016) as the main backbone in our experiments. For fair comparison, ABSGD is implemented with the same hyperparameters such as momentum parameter, initial step size, weight decay and step size decaying strategy, as the baseline momentum SGD method. For ABSGD, the moving average parameter $\gamma$ are tuned in $[0.1 : 0.1 : 1]$ by default. Without additional mentions, we directly use the results of baselines from their original papers by default.

**Datasets:** The original CIFAR-10 and CIFAR-100 data contain 50,000 training images and 10,000 validation with 10 and 100 classes, respectively. We construct the imbalanced version of training set of CIFAR10, CIFAR100 following the two strategies: Long-Tailed (LT) imbalance (Cao et al., 2019) and Step (ST) imbalance (Buda et al., 2018) with two different imbalance ratio $\rho = 10, \rho = 100$, and keep the testing set unchanged. The imbalance ratio $\rho$ is defined as the ratio between sample sizes of the most frequent and least frequent classes. The LT imbalance follows the exponentially decayed sample size between different categories. In ST imbalance, the number of examples are both equal within minority classes and majority classes but differs between the majority and minority classes. We denote the imbalanced versions of CIFAR10, CIFAR100 as CIFAR10-LT/ST, CIFAR100-LT/ST according the imbalanced strategies. ImageNet-LT (Liu et al., 2019) is a long-tailed subset of the original ImageNet-2012 by sampling a subset following the Pareto distribution with the power value 6. It has 115.8K images from 1000 categories, which include 4980 for the head class and 5 images for the tail class. The Places-LT dataset was also created by sampling from Places-2 (Zhou et al., 2017) using the same strategy as ImageNet-LT. It contains 62.5K training images from 365 classes with an imbalance ratio $\rho = 4980/5$. iNaturalist 2018 is a real world dataset whose class-frequency follows a heavy-tail distribution (iNatrualist, 2018). It contains 437K images from 8142 classes. The long-tail and step imbalance label distribution of the datasets are shown in Figure 2.

**Label-Distribution Independent Losses.** We first compare the effectiveness of our ABSGD method with standard momentum SGD method for DL. In particular, we consider two loss functions, cross-entropy (CE) loss and focal loss. The baseline method is the momentum SGD optimizing these losses, denoted by

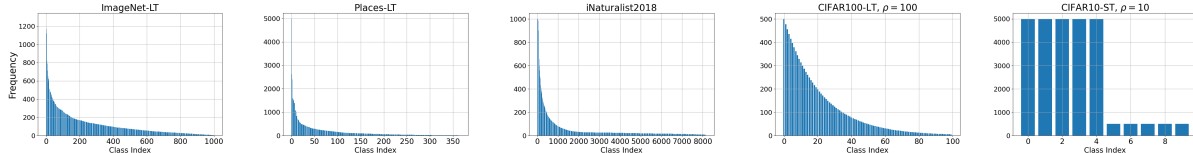

**Figure 2:** Long-tail label distributions of ImageNet-LT, Places-LT, iNaturalist2018 and CIFAR100 with imbalance ratio $\rho = 100$, and Step imbalance label distribution of CIFAR10 with imbalance ratio $\rho = 10$.

SGD (CE) and SGD (Focal). Our methods are denoted by ABSGD (CE) and ABSGD (Focal) that employ the two losses in our framework. This comparison is meaningful as in the online learning setting the prior knowledge of class-frequency is not known.

**Label-distribution Dependent Losses.** Next, we compare ABSGD with baseline methods that use label-distribution dependent losses. In particular, we consider class-balanced (CB) versions of three individual losses, including CE loss, focal loss, label-distribution-aware margin (LDAM) loss (Cao et al., 2019). The class-balanced weighing strategy is from (Cui et al., 2019), which uses the effective number of samples to define the weight. As a result, there are three categories of CB losses, i.e., CB-CE, CB-Focal, CB-LDAM. We use our method ABSGD with these different losses. In particular, ABSGD + CB-CE/Focal/LDAM uses a combination of class-level weighting and instance-level weighting, which is expected to have outstanding performance as it considers diversity between examples at both class level and individual level. For each of these losses, we consider two baseline optimization methods. The first method is the standard momentum SGD method with a practical useful trick (Cao et al., 2019) that defers adding the class-level weighting after a number of pre-training steps with no class-level weights to improve the performance. We denote the first method by SGD (XX), where XX denotes the loss function. The second method is the meta learning method (Jamal et al., 2020) that uses meta-learning on a separate validation data to learn individual weights and combines them with class-balanced weights. The meta learning method has been observed with SOTA results on these benchmark imbalanced datasets. We let META (XX) denote the second method. Our method is denoted by ABSGD (XX).

In the following, we compare ABSGD, SGD, and meta-learning methods by optimizing the same label-dependent and label-independent losses on imbalanced CIFAR datasets, and including more baselines on ImageNet-LT, Places-LT, and iNaturalist-LT.

**Table 1:** Top-1 testing accuracy (%), mean (std), of ResNet32 on imbalanced CIFAR-10 and CIFAR-100 trained with label-distribution independent losses. The results are reported over 3 independent runs.

| Dataset | Imbalance Type | long-tailed (LT) | | step (ST) | |
|---|---|---|---|---|---|
| | Imbalance Ratio | 100 | 10 | 100 | 10 |
| Cifar10 | SGD (CE) | 71.75 ($\pm$ 0.75) | 87.64 ($\pm$ 0.45) | 63.12 ($\pm$ 0.63) | 85.23 ($\pm$ 0.41) |
| | ABSGD (CE) | **72.43** ($\pm$ 0.31) | **87.93** ($\pm$ 0.25) | **66.24** ($\pm$ 0.35) | **85.84** ($\pm$ 0.27) |
| | SGD (Focal) | 70.86 ($\pm$ 0.68) | 87.10 ($\pm$ 0.41) | 63.31 ($\pm$ 0.61) | 85.55 ($\pm$ 0.46) |
| | ABSGD (Focal) | **72.48** ($\pm$ 0.28) | **87.26** ($\pm$ 0.35) | **65.03** ($\pm$ 0.33) | **85.67** ($\pm$ 0.30) |
| Cifar100 | SGD (CE) | 38.35 ($\pm$ 0.63) | 56.91 ($\pm$ 0.44) | 39.23 ($\pm$ 0.58) | 55.09 ($\pm$ 0.35) |
| | ABSGD (CE) | **39.77** ($\pm$ 0.34) | **57.44** ($\pm$ 0.25) | **39.76** ($\pm$ 0.37) | **55.15** ($\pm$ 0.29) |
| | SGD (Focal) | 39.05 ($\pm$ 0.71) | 56.89 ($\pm$ 0.41) | 39.32 ($\pm$ 0.61) | 54.45 ($\pm$ 0.43) |
| | ABSGD (Focal) | **39.37** ($\pm$ 0.38) | **57.08** ($\pm$ 0.29) | **39.75** ($\pm$ 0.39) | **55.40** ($\pm$ 0.33) |

## 4.1 Experimental Results on CIFAR Datasets

**Setups** Following the experimental setting in the literature, the initial learning rate is 0.1 and decays by a factor of 100 at the 160-th, 180-th epoch for both ABSGD and SGD in our experiments, respectively. The value of $\lambda$ in ABSGD tuned in $[1:1:10]$.

**Results.** We report the results with label independent losses in Table 1 and with label dependent losses in Table 2. We can see that ABSGD consistently outperforms SGD with a noticeable margin regardless

**Table 2:** Top-1 testing accuracy (%) of ResNet32 on imbalanced CIFAR-10 and CIFAR-100 trained with label-distribution dependent losses. The red numbers indicate the best in each category of class-weighted loss. The **bold red** numbers indicate the best in each imbalanced setting. The original paper of META does not include the results on the ST imbalanced setting, hence their missing results are marked by −.

| Datasets | Imbalanced CIFAR-10 | | | | Imbalanced CIFAR-100 | | | |
|---|---|---|---|---|---|---|---|---|
| Imbalance Type | long-tailed | | step | | long-tailed | | step | |
| Imbalance Ratio | 100 | 10 | 100 | 10 | 100 | 10 | 100 | 10 |
| Resampling (CE) | 71.78 | 86.99 | 61.16 | 84.59 | 38.87 | 56.92 | 38.84 | 54.35 |
| SGD (CB-CE) (Cui et al., 2019) | 72.37 | 86.77 | 61.84 | 83.80 | 38.70 | 57.56 | 21.31 | 53.39 |
| META (CB-CE) (Jamal et al., 2020) | 76.41 | **88.85** | - | - | 43.35 | 59.58 | - | - |
| ABSGD (CB-CE) | 79.34 | 88.57 | 72.93 | **88.42** | 45.54 | 61.12 | 45.89 | 60.77 |
| SGD (CB-Focal) (Cui et al., 2019) | 74.57 | 87.10 | 60.27 | 83.46 | 36.02 | 57.99 | 19.76 | 50.02 |
| META (CB-Focal)  (Jamal et al., 2020) | 78.90 | 88.37 | - | - | 44.70 | 59.59 | - | - |
| ABSGD (CB-Focal) | 79.53 | 88.76 | 76.33 | 85.90 | 44.11 | 59.14 | 45.41 | 59.75 |
| SGD (LDAM) (Cao et al., 2019) | 73.35 | 86.69 | 66.58 | 85.00 | 39.60 | 56.91 | 39.58 | 56.27 |
| SGD (CB-LDAM) (Cao et al., 2019) | 77.03 | 88.12 | 76.92 | 87.81 | 42.04 | 58.71 | 45.36 | 59.46 |
| META (CB-LDAM) (Jamal et al., 2020) | 80.00 | 87.40 | - | - | 44.08 | 58.80 | - | - |
| ABSGD (CB-LDAM) | **80.45** | 88.27 | **78.33** | 88.40 | 44.71 | 59.21 | 45.65 | 58.74 |

of imbalance strategies and imbalance ratio $\rho$. In particular, we have more than 2% improvements on the CIFAR10-ST and CIFAR100-LT, respectively with $\rho = 100$. For the label dependent losses, we have the following observations, comparing ABSGD with SGD, we can see that our method that incorporates the self-adaptive robust weighting scheme performs consistently better in all imbalanced settings. This verifies that the proposed self-adaptive weighting scheme is also effective even when applied on top of the class-level weighting strategy. It is notable that META requires a separate validation data and is more computationally expensive than our method. Hence, our method is a strong choice even compared with the SOTA meta learning method, especially for highly imbalanced tasks. Also, the improvements of ABSGD with CB losses over ABSGD with label independent losses verify the importance of prior label information in addressing the data imbalance problem.

## 4.2 Experimental Results on ImageNet-LT, Places-LT and iNaturalist2018

**Setups and baselines.** Next, we conduct experiments on large-scale datasets and compare ABSGD with more baselines. We conduct experiment on two different architectures, ResNet50 for ImageNet-LT and iNaturalist2018, and ResNet152 for Places-LT and iNaturalist2018. We compare ABSGD with several methods, which include single-stage methods such as momentum SGD for optimizing LDAM loss, CB-CE loss and CB-Focal loss, two-stage methods such as $\tau$-normalized (CB-CE), LWS (CB-CE) proposed in (Kang et al., 2019), and meta-learning method (META) (Jamal et al., 2020). For the two-stage decoupling strategy baseline methods (Kang et al., 2019), the first stage uses the standard uniform sampling to train the model with the CE loss, and the second stage fine tunes part of parameters in higher layers such as the fully connected (FC) layers and last block of (LB) feature layers. META also uses the two-stage strategy to improve the performance.

Here, to achieve the SOTA results, we investigate two-stage decoupling strategy for ABSGD. Hence, the two-stage decay $\lambda$ training scheme can be automatically applied. For ImageNet-LT, we jointly train the feature representation and classifier using momentum SGD in the first stage for 90 epochs from scratch, and finetune the FC layer for 90 epochs of ABSGD in the second stage. For Places-LT, we train the Last Block (LB) of the convolutions layer and Fully Connected (FC) layer for 90 epochs in the first stage using SGD with momentum from an ImageNet pretrained model, and finetune the FC and LB layer for 30 epochs in second stage using ABSGD. For iNaturalist2018, we run momentum SGD ($\beta = 0.9$) for 200 epochs in the first stage from the ImageNet pretrained model, and in the second stage we only finetune FC layer and LB of the neural network using ABSGD with $\lambda = 10$ for 30 epochs. $\lambda$ is tuned in $\{10, 20, 30\}$ for all datasets. The initial learning rates and learning scheme are described in Table 8 (Appendix). All of our results are reported based on 3 independent runs.

**Table 3:** Test top-1 accuracy(%) of different baseline methods on ImageNet-LT with Resnet50.

| Methods | Sampling | Loss | Stage-1 TV | Stage-2 TV | Results |
|---|---|---|---|---|---|
| Vanilla Model (Jamal et al., 2020) | None | CE | All | - | 41.0 |
| CB-CE  (Cui et al., 2019) | None | CE | All | - | 41.8 |
| Joint (Kang et al., 2019) | CB | CE | All | All | 41.6 |
| NCM (Kang et al., 2019) | CB | CE | All | FC | 44.3 |
| cRT (Kang et al., 2019) | CB | CE | All | FC | 43.3 |
| $\tau$-normalizer (Kang et al., 2019) | CB | CE | All | FC | 46.7 |
| META$^{\dagger}$ (Jamal et al., 2020) | None | CE | All | FC | 48.0 |
| ABSGD | None | CE | All | FC+LB | **48.2** |

**Table 4:** Test top-1 accuracy(%) of different baseline methods on Places-LT using ResNet50.

| Methods | Sampling | Loss | Stage-1 TV | Stage-2 TV | Results |
|---|---|---|---|---|---|
| Vanilla Model (Jamal et al., 2020) | - | CE | FC/LB+FC | - | 27.9/30.3 |
| Vanilla Model (Zhang et al., 2017) | - | Range | FC | - | 35.1 |
| Joint (Kang et al., 2019) | CB | CE | LB+FC | LB+FC | 30.2 |
| NCM (Kang et al., 2019) | CB | CE | LB+FC | FC | 36.3 |
| cRT (Kang et al., 2019) | CB | CE | LB+FC | FC | 36.7 |
| $\tau$-normalized (Kang et al., 2019) | None | CE | LB+FC | FC | 37.9 |
| OLTR$^{*}$  (Liu et al., 2019) | CB | CE | LB+FC | FC | 35.9 |
| META $^{\dagger}$ (Jamal et al., 2020) | None | CE | LB+FC | FC | 37.1 |
| ABSGD | None | CE | LB+FC | FC | **38.7** |

**Results** Table 3, 4  5 are the experimental results of ImageNet-LT, Places-LT and iNaturalist2018, respectively. To better understanding results, we make some notes in the table. *TV* represents Trainable Variable. *All* represents standard training process that optimizes all the parameters of the backbone. *FC* represents fully connected layer, LB represents the last block of feature layers in the backbone. The CB in the Sampling column denotes Class-Balanced Sampling (Cui et al., 2019) in the second stage. $^{\dagger}$ represents an additional balanced data set is required in the second stage. $^{*}$ denotes an additional memory is required in the second stage. The bold numbers and the numbers with underline in the table represent the best and the second best the numbers with underline on each dataset, respectively.

We can see that ABSGD combining with the two stage training strategy achieves best on all three datasets. For the ImageNet-LT dataset in Table 3, ABSGD has 0.2% improvements over the next best META method while has no requirements on the additional balanced validation datasets. For Places-LT, ABSGD has 0.9% improvements over than the best baseline, $\tau$-normalized. For the iNaturalist2018-LT in Table 5, ABSGD outperforms all other baselines by a large margin 0.3% and 0.6% for using both ResNet50 and ResNet152, respectively. To the best of our knowledge, 73.1% is the SOTA result on iNaturalist2018 dataset. In addition, it is worth to mention that all the other baselines takes the advantage of the Class-Balanced Sampling or additional balanced validation datasets (META), which makes ABSGD more favorable than the baselines.

### 4.3   Ablation Studies on CIFAR Datasets

In the ablation study, we first study ABSGD from different perspectives: a) the stagewise decay $\lambda$; b) the influence of the moving average parameter $\gamma$ on the testing accuracy. Then we plot the average instance robust weights for each class to show the attention of ABSGD towards the minority class.

**Two-stage decay of $\lambda$.**   To verify the model enjoys the benefits of stagewise decay $\lambda$ the same as the learning rate $\eta$, we compare the feature representations in both training and testing data between adopting the two-stage $\lambda$ decay training strategy and using a fixed value of $\lambda$ during the training. For two-stage strategy, we use $\lambda = 100$ in the first phase and decay it to 1 in the second phase. For fixed values of $\lambda$, we use $\lambda = 1$. The results are plotted in the second column of Figure 3 on CIFAR10-LT. It is clear to see using the stagewise strategy on $\lambda$ yields much better feature representations that are well separated between

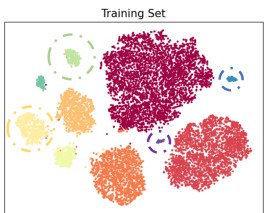 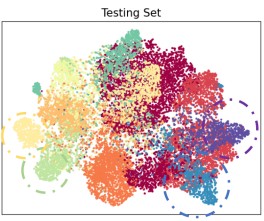 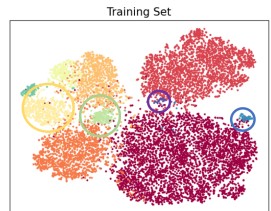 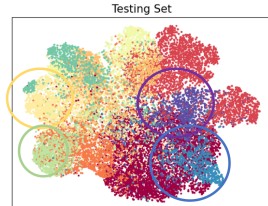

**Figure 3:** t-SNE visualization of feature representations of training & testing set on CIFAR10-LT ($\rho = 100$) with different $\lambda$ strategies. Left two figures: Two-stage decay of $\lambda$: first phase $\lambda = 100$ and second phase $\lambda = 1$. Right two figures: Fixed $\lambda = 1$.

**Table 5:** Top-1 testing accuracy(%) of different methods on iNaturalist2018 using ResNet50, ResNet152.

| Methods Network | Stage-2 TV | Results ResNet50 | Results ResNet152 |
|---|---|---|---|
| CE (Cui et al., 2019) | - | 65.8 | 69.0 |
| LDAM (Cao et al., 2019) | - | 68.0 | - |
| CB-Focal (Cui et al., 2019) | - | 61.1 | - |
| NCM (CE) (Kang et al., 2019) | FC | 63.1 | 67.3 |
| cRT (CB-CE) (Kang et al., 2019) | FC | 68.2 | 71.2 |
| $\tau$-Normalized (CE) (Kang et al., 2019) | FC | 69.3 | 72.5 |
| LWS (CB-CE) (Kang et al., 2019) | FC | 69.5 | 72.1 |
| META$^\dagger$ (CB-CE) (Jamal et al., 2020) | All | 67.6 | - |
| META$^\dagger$ (CB-Focal) (Jamal et al., 2020) | All | 67.7 | - |
| ABSGD (CB-CE) | FC | **69.8** | **73.1** |

different classes. In contrast, the learned feature representations with a fixed value $\lambda = 1$ are more cluttered. Thus the stagewise decay $\lambda$ strategy is better than using a fixed value of $\lambda$, which verifies our algorithmic choice. We also provide the convergence curves of different $\lambda$ strategies in Appendix.

**The sensitivity of the moving average parameter** $\gamma$   In the derivation of Theorem 1, $\gamma = O(\frac{1}{\sqrt{T}})$, which decreases to 0 when the total number of iterations increases. In practical training, we tune the $\gamma \in \{0.1, 0.3, 0.5, 0.7, 0.9\}$. We report the testing accuracy over 3 independent runs in Figure 4 (left two) and compare it with standard SGD training, the green dashed line. Here we can see that ABSGD achieves highest testing accuracy with $\gamma = 0.5$ on both CIFAR10-LT and CIFAR100-LT. All the results of ABSGD with different $\gamma$ are better or comparable than momentum SGD verifies the effectiveness of the moving average estimator Step 4 in Algorithm 1.

**The average instance weights per-class**   ABSGD is an instance-level weighting method. For each sample, ABSGD assigns a robust weight that is proportional to the scaled loss value. For ABSGD (CE), we plot the average robust weights for the samples in the minority and majority class in Figure 4 (right two). It can be clearly seen that the average weights of samples in minority class is greater than the average weights of samples in majority class, which verifies the intuition behind ABSGD.

## 5   Experimental Results on Label Noise Problem

To show the effectiveness of ABSGD for handling noisy labels, we provide empirical studies on the noisy label datasets in this section. We conduct experiments on CIFAR10, CIFAR100, and Clothing1M (Xiao et al., 2015) datasets. The noise rate is defined as the portion of samples whose ground truth label are randomly flipped. We follow the same setting as (Wang et al., 2019) and consider both the symmetric label noise and asymmetric label noise on CIFAR10 and CIFAR100 with the noisy rates $\{0.2, 0.4\}$ (Wang et al., 2019; Patrini et al., 2017; Zhang & Sabuncu, 2018) in our experiments. The symmetric (uniform) noisy labels are

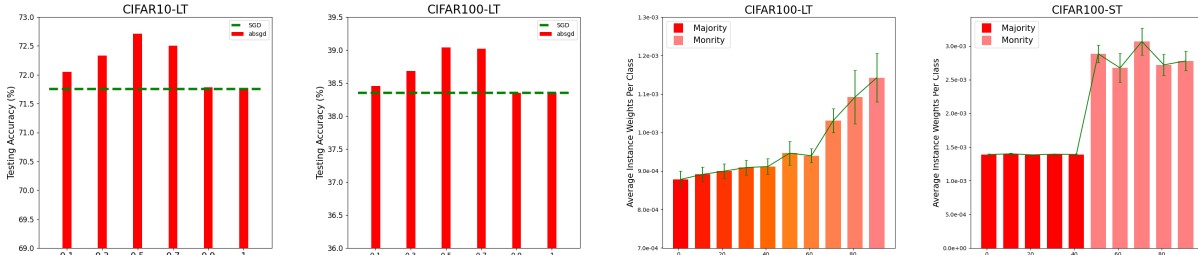

**Figure 4:** Left two: the influence of $\gamma$ on the CIFAR10-LT and CIFAR100-LT with imbalance ratio 100. The results are reported over 3 independent runs. The green error bar is the stand deviation of each results. Right two: the average instance weights for difference classes during the training process on CIFAR100-LT and CIFAR100-ST with imbalanced ratio 100 on ResNet32.

generated by flipping the labels of a given proportion of training samples to one of the other class labels uniformly. The asymmetric noisy labels are class-dependent noise, in which the flipping of labels only occur within a specific set of classes. Please refer to the Noise setting section in (Wang et al., 2019) for details. The Clothing1M is a real-world large-scale label noisy dataset and includes 14 classes with 1M training images in total.

**Baselines** We compare ABSGD with SGD and a mini-batch based method for solving the min-min DRO formulation (4) named EG (Majidi et al., 2021) and CIW with $\alpha = 1$ (Kumar & Amid, 2021) with different losses. The first is the standard CE loss. Then a theoretically grounded generalized cross entropy loss, named as TCE, has been proposed later on (Zhang & Sabuncu, 2018). Furthermore, (Wang et al., 2019) proposed a symmetric loss, named SCE, to address the under learning and overfitting problem that widely exists in the noisy labels. For crafting loss hyperparameters, we tune the symmetric parameters in SCE $\alpha, \beta \in \{0.1, 1, 0.5, 1, 5\}$ and the truncated parameter $q$ in TCE is tuned in $\{0.1, 0.5, 0.7\}$. The momentum parameter $\gamma$ for ABSGD is tuned in $\{0.1, 0.5, 0.9\}$.

### 5.1 Experimental Results on CIFAR Datasets

**Experimental Setting.** Following the setting in (Wang et al., 2019), we use a 4-layer CNN proposed in (Wang et al., 2019) for CIFAR10 data. For the CIFAR100, we use ResNet18 for the symmetric noisy labels and the asymmetric noisy labels. We report the results of using CE and TCE losses optimized by SGD, and SCE optimized by SGD, EG, CIW and ABSGD, respectively. The weight decay for different methods are tuned in $\{1e\text{-}4, 5e\text{-}4, 1e\text{-}3, 5e\text{-}3\}$. We train the model for 120 epochs and the batch size is fixed as 128 for all settings. The initial learning rates are tuned in $\{1e\text{-}3, 1e\text{-}2, 1e\text{-}1\}$ and decayed at the epoch of 40, and 80 epochs by a factor of 10. The ABSGD hyper-parameter $\lambda$ is tuned in $\{-0.1, -0.5, -1, -2, -3\}$.

**Results.** The results are reported in Table 6. Among the three baselines, SCE achieves better/comparable experimental results in most of the different models, settings and datasets. Then the testing accuracy is consistently improved further by optimizing SCE with the proposed ABSGD. By comparing the results across different noisy rates, we can see that our ABSGD(SCE) improves more when the noisy rate increases.

### 5.2 Experimental Results on Clothing1M

**Experimental Setting.** We train the ResNet50 starting from the ImageNet pretrained model for all the baselines following the same setting as (Wang et al., 2019) on the Clothing1M dataset. The training phase includes 10 epochs, and the initial learning rate is fixed as 0.002 and decayed by a factor of 10 at the 5th epoch for all the methods. The weight decay is set as $1e\text{-}2$. The $\lambda$ for ABSGD is tuned in $\{-1, -5, -10, -15\}$. We report the results on CE, TCE, SCE optimized by standard SGD, EG and ABSGD, respectively.

**Results.** The results are reported in Table 7. We can see that ABSGD has better testing accuracy than SGD and EG for all losses.

**Table 6:** Top-1 testing accuracy (%) on noisy labelled CIFAR10 and CIFAR100 data of different methods. Results are reported over 5 independent runs. Bold and underline represent the best and second results

| | | Symmetric | | Asymmetric | |
|---|---|---|---|---|---|
| | Noisy Rate | 0.2 | 0.4 | 0.2 | 0.4 |
| CIFAR10 | SGD(CE) | 88.59 ($\pm$ 0.21) | 85.75 ($\pm$ 0.31) | 86.62 ($\pm$ 0.27) | 80.81 ($\pm$ 0.29) |
| | SGD(TCE) | 89.87 ($\pm$ 0.27) | 86.84 ($\pm$ 0.32) | 88.97 ($\pm$ 0.31) | 80.85 ($\pm$ 0.27) |
| | SGD(SCE) | 90.05 ($\pm$ 0.23) | 87.83 ($\pm$ 0.33) | 90.25 ($\pm$ 0.34) | 81.91 ($\pm$ 0.42) |
| | EG(SCE) | 90.25 ($\pm$ 0.21) | 88.13 ($\pm$ 0.29) | 90.55 ($\pm$ 0.32) | 84.47($\pm$ 0.25) |
| | CIW(CE) | 90.29 ($\pm$ 0.23) | 87.92 ($\pm$ 0.19) | 88.99 ($\pm$ 0.24) | 87.18 ($\pm$ 0.21) |
| | CIW(SCE) | 90.79 ($\pm$ 0.25) | 88.21 ($\pm$ 0.22) | 89.97 ($\pm$ 0.23) | 85.13 ($\pm$ 0.19) |
| | ABSGD(CE) | 90.64 ($\pm$ 0.20) | 88.31 ($\pm$ 0.19) | 90.15 ($\pm$ 0.22) | **87.84** ($\pm$ 0.25) |
| | ABSGD(SCE) | **91.15** ($\pm$ 0.18) | **88.65** ($\pm$ 0.21) | **91.04** ($\pm$ 0.21) | 86.10 ($\pm$ 0.21) |
| CIFAR100 | SGD(CE) | 68.21 ($\pm$ 0.27) | 62.54($\pm$ 0.22) | 69.57 ($\pm$ 0.32) | 62.93 ($\pm$ 0.28) |
| | SGD(SCE) | 68.28 ($\pm$ 0.29) | 60.72 ($\pm$ 0.23) | 69.31 ($\pm$ 0.31) | 64.22 ($\pm$ 0.21) |
| | SGD(TCE) | 65.12 ($\pm$ 0.39) | 59.61 ($\pm$ 0.32) | 67.98($\pm$ 0.28) | 60.88 ($\pm$ 0.27) |
| | EG(SCE) | 69.53 ($\pm$ 0.21) | 65.36 ($\pm$ 0.19) | 69.61 ($\pm$ 0.23) | 67.01 ($\pm$ 0.24) |
| | CIW(CE) | 70.21 ($\pm$ 0.20) | 65.89 ($\pm$ 0.19) | 69.29 ($\pm$ 0.22) | 64.75 ($\pm$ 0.23) |
| | CIW(SCE) | 69.53 ($\pm$ 0.19) | 65.38 ($\pm$ 0.23) | 70.07 ($\pm$ 0.21) | 67.19 ($\pm$ 0.24) |
| | ABSGD(CE) | 70.63 ($\pm$ 0.19) | 66.23 ($\pm$ 0.24) | **70.70** ($\pm$ 0.21) | **68.16** ($\pm$ 0.22) |
| | ABSGD(SCE) | **71.23** ($\pm$ 0.19) | **66.39** ($\pm$ 0.20) | 69.98 ($\pm$ 0.23) | 65.26 ($\pm$ 0.24) |

**Table 7:** Top-1 testing accuracy (%) on Clothing1M data of different methods. Results are reported over 3 independent runs.

| loss | SGD | EG | CIW | ABSGD |
|---|---|---|---|---|
| CE | 69.05 ($\pm$ 0.21) | 69.42 ($\pm$ 0.20) | 69.53 ($\pm$ 0.31) | **69.79** ($\pm$ 0.18) |
| SCE | 69.31 ($\pm$ 0.31) | 69.32 ($\pm$ 0.21) | 69.22 ($\pm$ 0.29) | **69.93** ($\pm$ 0.11) |
| TCE | 68.28 ($\pm$ 0.23) | 68.66 ($\pm$ 0.19) | 68.51 ($\pm$ 0.33) | **68.69** ($\pm$ 0.18) |

# 6 Conclusion

In this paper, we propose a unified framework, ABSGD, for addressing the data imbalance and noisy label problem. We provide the theoretical analysis both for the SGD-style and Adam-style updates. Empirical studies on multiple benchmark datasets with different models show the outstanding performance of ABSGD compared with several strong baselines.

**Acknowledgments**

Q. Qi and T. Yang are partially supported by NSF Career Award #1844403, NSF Grant #2110545, and NSF-Amazon Joint Grant #2147253. Part of this work was supported by Alibaba Gift funding. T. Yang supervised the work including formulations, analysis and experiments, and helped write the paper. Q. Qi is responsible for conducting experiments, writing proofs and papers. Y. Xu, W. Yin and R. Jin provided suggestions in the early stage of this work. The work of Y. Xu and R. Jin were done when they were at Alibaba Group at Seattle.

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

## 7 Appendix

**Table 8:** General hyperparameter settings in different experiments of section 4

| Datasets | Initial Step Size | Weight Decay | Schedule | Batch Size | Momentum |
|---|---|---|---|---|---|
| CIFAR10-ST/LT | 0.1 | 2e-4 | Stagewise Decay (Yuan et al., 2019) | 128 | 0.9 |
| CIFAR100-ST/LT | 0.1 | 2e-4 | Stagewise Decay (Yuan et al., 2019) | 128 | 0.9 |
| ImageNet-LT | 0.05 | 5e-4 | Cosine Annealing (Loshchilov & Hutter, 2016) | 512 | 0.9 |
| Places-LT | 0.05 | 5e-4 | Cosine Annealing (Loshchilov & Hutter, 2016) | 512 | 0.9 |
| iNaturalist2018 | 0.2 | 1e-4 | Cosine Annealing (Loshchilov & Hutter, 2016) | 512 | 0.9 |

**The benefits of momentum** Next, we verify that adding the momentum term can dramatically improve performance. The results are plotted in the left two columns of Figure (5) on CIFAR10-LT ($\rho = 100$) and CIFAR100-LT ($\rho = 100$) datasets, where we plot the testing accuracy vs the epochs of optimization with an average of 3 runs. The results clearly show that including a momentum term helps improve performance and stabilize the training process.

**The Effect of Damping $\lambda$ on Convergence.** Figure 3 shows the advantage of using the damping strategy on $\lambda$ on feature representation learning. Here, we plot the convergence curves in terms of testing accuracy in Figure 5. It is obvious to see that damping $\lambda$ achieves higher test accuracy over fixed values of $\lambda$, which also verifies our choice of damping $\lambda$.

**Running Time Comparison** To show the efficiency of ABSGD, we conduct an experiment on CIFAR-10 data with different networks on NVIDIA GeForce GTX 1080 Ti. The running time per iteration (seconds) of SGD, ABSGD and META methods are shown in the following table. It is clear to see that ABSGD has a comparable running time as SGD, while the per iteration running time of META is way slower than SGD and ABSGD.

**Table 9:** Tunning time (seconds) per iterations of SGD, ABSGD, and META (Jamal et al., 2020) methods on CIFAR-10 dataset with different networks.

| Network(# Param.) | SGD | ABSGD | META |
|---|---|---|---|
| ResNet32 (0.46M) | 0.0167 | 0.0176 | 0.376 |
| ResNet44 (0.44M) | 0.0234 | 0.0250 | 0.474 |
| ResNet56 (0.85M) | 0.0284 | 0.0296 | 0.566 |
| ResNet110 (1.7M) | 0.0684 | 0.0692 | 0.882 |

**Experiements on Convmixer** We have implemented the newly proposed structure, named as Convmixer (Trockman & Kolter, 2022), which operates convolutional layers on small patches. Convmixer has been shown to achieve competitive results as ViT models but with faster training speed and fewer parameters. We conducted an experiment by comparing ABSGD with CE loss and SGD for optimizing CE loss with on CIFAR10 dataset in the long tail setting with an imbalance ratio of 10, and 100. The result is presented in Table 10.

**Table 10:** Empirical Results on imbalanced dataset CIFAR10-LT with Convmixer structure

| Imbalance Ratio | SGD (CE) | ABSGD (CE) |
|---|---|---|
| 10 | 82.1 ($\pm$ 0.28) | 83.90 ($\pm$ 0.23) |
| 100 | 63.2 ($\pm$ 0.31) | 66.31 ($\pm$ 0.21) |

**Comparison with optimal sample complexity algorithm with RECOVER**

In this section, we compare ABSGD with RECOVER on the CIFAR dataset, which achieves optimal sample complexity when solving regularized DRO with KL-divergence. The results are reported in Table 11. We can see that ABSGD achieve better empirical that RECOVER on imbalanced datasets.

**Table 11:** Experimental results on imbalanced CIFAT10-ST, CIFAR100-ST on ResNet32

|  | Imbalance Ratio | ABSGD | RECOVER |
|---|---|---|---|
| CIFAR10-ST | 10 | 85.84 ($\pm$ 0.27) | 82.61 ($\pm$ 0.43) |
|  | 100 | 66.24 ($\pm$ 0.35) | 63.53 ($\pm$ 0.71) |
| CIFAR100-ST | 10 | 55.15 ($\pm$ 0.29) | 52.52 ($\pm$ 0.61) |
|  | 100 | 39.76 ($\pm$ 0.37) | 37.21 ($\pm$ 0.64) |

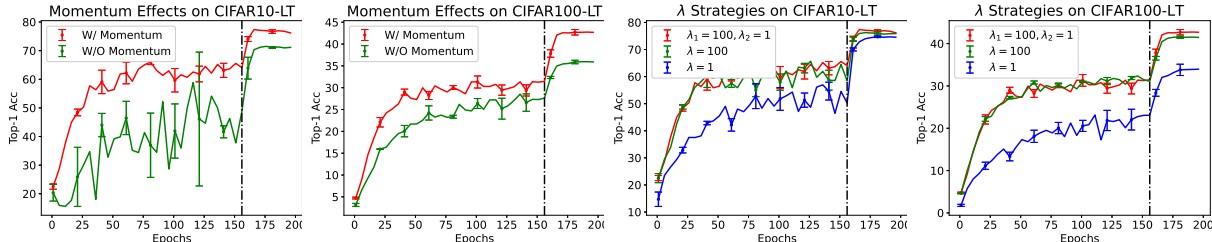

**Figure 5:** Ablation studies on CIFAR10-LT and CIFAR100-LT datasets: Left two images: comparing ABSGD with (W/) momentum vs without(W/O) momentum. Right two images: comparing ABSGD with different $\lambda$ strategies on CIFAR-LT datasets. The black dashed lines represent the epoch where the learning rates are decayed. For the red line, $\lambda$ also decays from $\lambda_1$ to $\lambda_2$ at the dashed line epoch. The results are averaged over 3 random trials.

# 8 Theoretical Analysis of Theorem 1

**Notations** Denote $f(s) = \lambda \log(s)$, $g(\mathbf{w}; \mathbf{z}) = \exp(\frac{L(\mathbf{w};\mathbf{z})}{\lambda})$ and $g(\mathbf{w}) = \mathbb{E}_{\mathbf{z}}[g(\mathbf{w}; \mathbf{z})]$, then

$$F_\lambda(\mathbf{w}) = f(g(\mathbf{w})) = f(\mathbb{E}_{\mathbf{z}}[g(\mathbf{w}; \mathbf{z})])$$

And $L_g$, $L_f$, $C_g$, $C_f$, $D_s$, $D_{\mathcal{G}}$ are positive constants. By denoting $L_g = \frac{C_0 L_l}{\lambda} + \frac{C_0 C_1}{\lambda^2}$, $C_g = \frac{C_0 \sqrt{C_1}}{\lambda}$, $L_f = \lambda$, and $C_f = \lambda$, we first derive the smooth and continuous property of $f(\cdot)$ and $g(\mathbf{w}; \mathbf{z})$ for $\forall \mathbf{z} \sim \mathcal{D}$ implied by Assumption 1 with the following propositions.

**Proposition 1.** $g(\mathbf{w})$ *is a $L_g$-smooth and $C_g$-Lipschitz continuous function.*

*Proof.* By Assumption 1 and Theorem 1, $\|\nabla L(\mathbf{w}; \mathbf{z}) - \nabla L(\mathbf{w}'; \mathbf{z})\| \le L_l \|\mathbf{w} - \mathbf{w}'\|, \forall \mathbf{w}, \mathbf{w}'$, $g(\mathbf{w}; \mathbf{z}) = \exp(\frac{L(\mathbf{w};\mathbf{z})}{\lambda}) \le C_0$, $L(\mathbf{w}; \mathbf{z}) \ge 0$ and $\|\nabla L(\mathbf{w}; \mathbf{z})\|^2 \le C_1$, we have $1 \le g(\mathbf{w}; \mathbf{z}) \le C_0, \forall \mathbf{z} \sim \mathcal{D}$ and

$$
\begin{aligned}
\|\nabla^2 g(\mathbf{w})\| &= \|\frac{1}{n}\sum_{i=1}^n \nabla^2 g(\mathbf{w}; \mathbf{z}_i)\| \le \frac{1}{n}\sum_{i=1}^n \|\nabla^2 g(\mathbf{w}; \mathbf{z}_i)\| \\
&= \mathbb{E}_{\mathbf{z}}[\|\nabla^2 g(\mathbf{w}; \mathbf{z})\|] = \mathbb{E}_{\mathbf{z}}[\frac{1}{\lambda}\|\nabla^2 L(\mathbf{w}, \mathbf{z})\exp(\frac{L(\mathbf{w};\mathbf{z})}{\lambda}) + \exp(\frac{L(\mathbf{w};\mathbf{z})}{\lambda})\frac{\nabla L(\mathbf{w};\mathbf{z})}{\lambda}\nabla L(\mathbf{w}; \mathbf{z})^\top\|] \\
&\le \mathbb{E}_{\mathbf{z}}[\frac{1}{\lambda}\|\nabla^2 L(\mathbf{w}; \mathbf{z})\exp(\frac{L(\mathbf{w};\mathbf{z})}{\lambda})\| + \frac{1}{\lambda^2}\|\exp(\frac{L(\mathbf{w};\mathbf{z})}{\lambda})\nabla L(\mathbf{w};\mathbf{z})\nabla L(\mathbf{w};\mathbf{z})^\top\|] \\
&\le \frac{C_0 L_l}{\lambda} + \frac{C_0}{\lambda^2}\mathbb{E}_{\mathbf{z}}[\|\nabla L(\mathbf{w};\mathbf{z})\|^2] \le \frac{C_0 L_l}{\lambda} + \frac{C_0 C_1}{\lambda^2}
\end{aligned}
\tag{10}
$$

In addition, with the assumption in Theorem 1,

$$
\begin{aligned}
\|\nabla g(\mathbf{w})\| &= \|\mathbb{E}_{\mathbf{z}}[\nabla g(\mathbf{w};\mathbf{z})]\| \le \mathbb{E}_{\mathbf{z}}[\|\nabla g(\mathbf{w};\mathbf{z})\|] = \frac{1}{\lambda}\mathbb{E}_{\mathbf{z}}[\|\nabla L(\mathbf{w};\mathbf{z})\exp(\frac{L(\mathbf{w};\mathbf{z})}{\lambda})\|] \\
&\le \frac{C_0}{\lambda}\mathbb{E}_{\mathbf{z}}[\|\nabla L(\mathbf{w};\mathbf{z})\|] \le \frac{C_0}{\lambda}\sqrt{\mathbb{E}_{\mathbf{z}}[\|\nabla L(\mathbf{w};\mathbf{z})\|^2]} \le \frac{C_0 \sqrt{C_1}}{\lambda}
\end{aligned}
\tag{11}
$$

$\square$

**Proposition 2.** $f(s) = \lambda \log(s)$ *is a $L_f$-smooth and $C_f$-Lipschitz continuous function.*

*Proof.* $\nabla f(s) = \frac{\lambda}{s}$. As $s = g(\mathbf{w}; \mathbf{z}) \in (1, C_0]$, $\nabla f(s) \leq \lambda$, which implies $\|\nabla f(s)\| \leq \lambda$. In addition,

$$\|\nabla f(s_1) - \nabla f(s_2)\| = \|\frac{\lambda}{s_1} - \frac{\lambda}{s_2}\| \leq \left\|\frac{\lambda}{s_1 s_2}\right\| \|s_1 - s_2\| < \lambda \|s_1 - s_2\|. \tag{12}$$

$\square$

Following the assumption 1, Proposition 1 and 2, and the conditions in Theorem 1, then by let $G = \max(C_f^2, C_g^2)$, $L_F = L_f$ the following inequalities hold:

- $\|\nabla g(\mathbf{w}; \mathbf{z})\|^2 \leq G, \forall \mathbf{z}, |\nabla f(s)|^2 \leq G$

- $\mathbb{E}[|g(\mathbf{w}; \mathbf{z}) - g(\mathbf{w})|^2] \leq V_g$

- $F(\mathbf{w})$ is $L_F$ smooth.

Next, we provide the following lemma to describe the objective gap between adjacent solutions for any $L_F$-smooth function $F(\mathbf{w}) : \mathbb{R}^d \to \mathbb{R}$ with the $\mathbf{w}_{t+1} = \mathbf{w}_t - \tilde{\eta}\mathbf{v}_{t+1}$ update. $\mathbf{v}_t \in \mathbb{R}^d$ can be any vector.

**Lemma 1.** *Consider a sequence update $\mathbf{w}_{t+1} = \mathbf{w}_t - \tilde{\eta}\mathbf{v}_{t+1}$, suppose $c_l\eta \leq \tilde{\eta} \leq c_u\eta$ for a $L_F$-smooth function $F$, with $\eta L_F \leq c_l/2c_u^2$ we have*

$$F(\mathbf{w}_{t+1}) \leq F(\mathbf{w}_t) + \frac{\eta}{2}\|\nabla F(\mathbf{w}_t) - \mathbf{v}_{t+1}\|^2 - \frac{\eta}{2}\|\nabla F(\mathbf{w}_t)\|^2 - \frac{\eta}{4}\|\mathbf{v}_{t+1}\|^2.$$

*Proof.* Due the smoothness of $F$, we can prove that under $\eta L_F \leq c_l/2c_u^2$

$$F(\mathbf{w}_{t+1}) \leq F(\mathbf{w}_t) + \nabla F(\mathbf{w}_t)^\top(\mathbf{w}_{t+1} - \mathbf{w}_t) + \frac{L_F}{2}\|\mathbf{w}_{t+1} - \mathbf{w}_t\|^2$$

$$= F(\mathbf{w}_t) - \tilde{\eta}\nabla F(\mathbf{w}_t)^\top\mathbf{v}_{t+1} + \frac{L_F\tilde{\eta}^2}{2}\|\mathbf{v}_{t+1}\|^2$$

$$\leq F(\mathbf{w}_t) + \frac{\tilde{\eta}}{2}\|\nabla F(\mathbf{w}_t) - \mathbf{v}_{t+1}\|^2 - \frac{\tilde{\eta}c_l}{2}\|\nabla F(\mathbf{w}_t)\|^2 + (\frac{L_F\tilde{\eta}^2}{2} - \frac{\tilde{\eta}}{2})\|\mathbf{v}_{t+1}\|^2$$

$$\leq F(\mathbf{w}_t) + \frac{\eta c_u}{2}\|\nabla F(\mathbf{w}_t) - \mathbf{v}_{t+1}\|^2 - \frac{\eta c_l}{2}\|\nabla F(\mathbf{w}_t)\|^2 + (\frac{L_F\eta^2 c_u^2}{2} - \frac{\eta c_l}{2})\|\mathbf{v}_{t+1}\|^2$$

$$\leq F(\mathbf{w}_t) + \frac{\eta c_u}{2}\|\nabla F(\mathbf{w}_t) - \mathbf{v}_{t+1}\|^2 - \frac{\eta c_l}{2}\|\nabla F(\mathbf{w}_t)\|^2 - \frac{\eta c_l}{4}\|\mathbf{v}_{t+1}\|^2$$

$\square$

**Remark** When $\mathbf{v}_{t+1}$ is a stochastic gradient estimator, $\|\nabla F(\mathbf{w}) - \mathbf{v}_{t+1}\|^2$ represents the stochastic gradient estimator variance. In the following, we show that $\|\nabla F(\mathbf{w}) - \mathbf{v}_{t+1}\|^2$ is decreasing for the proposed stochastic estimator in ABSGD and ABAdam, which guarantees the convergence of the algorithms.

Next, the next lemma describes track the $\|\nabla F(\mathbf{w}) - \mathbf{v}_{t+1}\|^2$

**Lemma 2.** *Suppose Assumption 1 holds, then with the updates of $s_{t+1} = (1-\gamma)s_t + \gamma g(\mathbf{w}_t; \mathbf{z}_t)$ $\mathbf{v}_{t+1} = (1-\tilde{\beta})\mathbf{v}_t + \tilde{\beta}\nabla g(\mathbf{w}_t; \mathbf{z}_t)\nabla f(s_{t+1})$, for $\forall t > 0$, the following inequality holds:*

$$\mathbb{E}_t\|\nabla F(\mathbf{w}_t) - \mathbf{v}_{t+1}\|^2$$

$$\leq (1-\tilde{\beta})\|\nabla F(\mathbf{w}_{t-1}) - \mathbf{v}_t\|^2 + \frac{4}{\beta_0}L_F^2\|\mathbf{w}_t - \mathbf{w}_{t-1}\|^2 + 5\tilde{\beta}G\mathbb{E}_t\|g(\mathbf{w}_t) - s_{t+1}\|^2 + 4\tilde{\beta}^2G^2$$

The third term on the right hand side can be bounded with the following lemma:

**Lemma 3** (Lemma 2, wang2017stochastic). *Consider a moving average sequence $s_{t+1} = (1-\gamma)s_t + \gamma g(\mathbf{w}_t; \mathbf{z}_t)$ for tracking $g(\mathbf{w}_t)$, where $\mathbb{E}_\mathbf{z}[g(\mathbf{w}_t; \mathbf{z})] = g(\mathbf{w}_t)$ and $g$ is a $C_g$-Lipchitz continuous operator. Then we have*

$$\mathbb{E}_t[|s_{t+1} - g(\mathbf{x}_t)|^2] \leq (1-\gamma)|s_t - g(\mathbf{x}_{t-1})|^2 + \gamma^2\mathbb{E}_t[\|g(\mathbf{x}_t; \mathbf{z}_t) - g(\mathbf{x}_t)\|^2] + \frac{2L^2\|\mathbf{w}_t - \mathbf{w}_{t-1}\|^2}{\gamma}.$$

*Proof of Lemma 2.*

$$\|\nabla F(\mathbf{w}_t) - \mathbf{v}_{t+1}\|^2$$
$$= \|(1-\tilde{\beta})\nabla F(\mathbf{w}_{t-1}) - (1-\tilde{\beta})\mathbf{v}_t - \tilde{\beta}\nabla g(\mathbf{w}_t; \mathbf{z}_t)\nabla f(s_{t+1}) + \nabla F(\mathbf{w}_t) - (1-\tilde{\beta})\nabla F(\mathbf{w}_{t-1})\|^2$$
$$\leq \|(1-\tilde{\beta})(\nabla F(\mathbf{w}_{t-1}) - \mathbf{v}_t) + \tilde{\beta}\nabla g(\mathbf{w}_t; \mathbf{z}_t)\nabla f(g(\mathbf{w}_t)) - \nabla g(\mathbf{w}_t; \mathbf{z}_t)\nabla f(s_{t+1}))$$
$$+ (1-\tilde{\beta})(\nabla F(\mathbf{w}_t) - \nabla F(\mathbf{w}_{t-1})) + \tilde{\beta}(\nabla F(\mathbf{w}_t) - \nabla g(\mathbf{w}_t; \mathbf{z}_t)\nabla f(g(\mathbf{w}_t)))\|^2$$

Taking expectation on both sides over $\mathbf{z}_t$ conditioned on historical randomness and noting that $\mathbb{E}_t[\nabla F(\mathbf{w}_t) - \nabla g(\mathbf{w}_t; \mathbf{z}_t)\nabla f(g(\mathbf{w}_t))] = 0$, we have

$$\mathbb{E}_t\|\nabla F(\mathbf{w}_t) - \mathbf{v}_{t+1}\|^2$$
$$\leq \mathbb{E}_t\|(1-\tilde{\beta})(\nabla F(\mathbf{w}_{t-1}) - \mathbf{v}_t) + (1-\tilde{\beta})(\nabla F(\mathbf{w}_t) - \nabla F(\mathbf{w}_{t-1}))$$
$$+ \tilde{\beta}(\nabla g(\mathbf{w}_t; \mathbf{z}_t)\nabla f(g(\mathbf{w}_t)) - \nabla g(\mathbf{w}_t; \mathbf{z}_t)\nabla f(s_{t+1}))\|^2$$
$$+ \tilde{\beta}^2\mathbb{E}_t\|\nabla F(\mathbf{w}_t) - \nabla g(\mathbf{w}_t; \mathbf{z}_t)\nabla f(g(\mathbf{w}_t))\|^2$$
$$+ 2\tilde{\beta}^2\mathbb{E}_t[\|\nabla g(\mathbf{w}_t; \mathbf{z}_t)\nabla f(g(\mathbf{w}_t)) - \nabla g(\mathbf{w}_t; \mathbf{z}_t)\nabla f(s_{t+1}))\|\|\nabla F(\mathbf{w}_t) - \nabla g(\mathbf{w}_t; \mathbf{z}_t)\nabla f(g(\mathbf{w}_t))\|]$$
$$\overset{(a)}{\leq} (1-\tilde{\beta})\|\nabla F(\mathbf{w}_{t-1}) - \mathbf{v}_t\|^2 + 2(1+\frac{1}{\tilde{\beta}})(1-\tilde{\beta})^2 L_F^2\|\mathbf{w}_t - \mathbf{w}_{t-1}\|^2 + 2\beta_0^2(1+\frac{1}{\tilde{\beta}})G\mathbb{E}_t\|g(\mathbf{w}_t) - s_{t+1}\|^2$$
$$+ 2\tilde{\beta}^2\|\nabla F(\mathbf{w}_t) - \nabla g(\mathbf{w}_t; \mathbf{z}_t)\nabla f(g(\mathbf{w}_t))\|^2 + \tilde{\beta}^2 G\mathbb{E}_t\|g(\mathbf{w}_t) - s_{t+1}\|^2$$
$$\overset{(b)}{\leq} (1-\tilde{\beta})\|\nabla F(\mathbf{w}_{t-1}) - \mathbf{v}_t\|^2 + \frac{4}{\tilde{\beta}}L_F^2\|\mathbf{w}_t - \mathbf{w}_{t-1}\|^2 + 4\tilde{\beta}G\mathbb{E}_t\|g(\mathbf{w}_t) - s_{t+1}\|^2$$
$$+ 4\tilde{\beta}^2 G^2 + \tilde{\beta}^2 G\mathbb{E}_t\|g(\mathbf{w}_t) - s_{t+1}\|^2$$
$$\overset{(c)}{\leq} (1-\tilde{\beta})\|\nabla F(\mathbf{w}_{t-1}) - \mathbf{v}_t\|^2 + \frac{4}{\tilde{\beta}}L_F^2\|\mathbf{w}_t - \mathbf{w}_{t-1}\|^2 + 5\tilde{\beta}G\mathbb{E}_t\|g(\mathbf{w}_t) - s_{t+1}\|^2$$
$$+ 4\tilde{\beta}^2 G^2$$

where the inequality $(a)$ is due to $\|a+b\|^2 \leq (1+\tilde{\beta})\|a\|^2 + (1+\frac{1}{\tilde{\beta}})\|b\|^2$, and $ab \leq \frac{a^2}{2} + \frac{b^2}{2}$. The inequality $(b)$ applies $\tilde{\beta} \leq 1, 1+\frac{1}{\tilde{\beta}} \leq \frac{2}{\tilde{\beta}}$, $\mathbb{E}_t\|\nabla F(\mathbf{w}_t) - \nabla g(\mathbf{w}_t; \mathbf{z}_t)\nabla f(g(\mathbf{w}_t))\|^2 = \|\nabla f(g(\mathbf{w}_t))\|^2\mathbb{E}_t[\|\nabla g(\mathbf{w}_t) - \nabla g(\mathbf{w}_t; \mathbf{z}_t)\|^2] \leq \|\nabla f(g(\mathbf{w}_t))\|^2\mathbb{E}_t[\|\nabla g(\mathbf{w}_t)\|^2 - \|\nabla g(\mathbf{w}_t; \mathbf{z}_t)\|^2] \leq \|\nabla f(g(\mathbf{w}_t))\|^2\mathbb{E}_t[\|\nabla g(\mathbf{w}_t)\|^2 + \|\nabla g(\mathbf{w}_t; \mathbf{z}_t)\|^2] \leq 2G^2$ and the last inequality $(c)$ is due to $\tilde{\beta}G \geq \tilde{\beta}^2 G$.

$\square$

## 8.1 Convergence Analysis of ABSGD

Without loss of generality, we ignore $r$ and consider the objective in the form of $F(\mathbf{w}) = f(g(\mathbf{w}))$, where $f = \lambda\log(\cdot)$ is a deterministic function and $g = \mathbb{E}[\exp(L(\mathbf{w}; \mathbf{z})/\lambda)]$ is a stochastic function, and at each iteration, we only sample one data $\mathbf{z}_t$ for evaluating $g(\mathbf{w}_t; \mathbf{z}_t)$ and $\nabla g(\mathbf{w}_t; \mathbf{z}_t)$. To provide the convergence analysis for ABSGD, the updates of Steps 3-6 in Algorithm 1 can be equivalently written as:

$$s_{t+1} = (1-\gamma)s_t + \gamma g(\mathbf{w}_t; \mathbf{z}_t)$$
$$\mathbf{v}_{t+1} = \beta\mathbf{v}_t - \eta\nabla g(\mathbf{w}_t; \mathbf{z}_t)\nabla f(s_{t+1})$$
$$\mathbf{w}_{t+1} = \mathbf{w}_t + \mathbf{v}_{t+1}$$

With some change of variable, the above update is equivalent to

$$s_{t+1} = (1 - \gamma)s_t + \gamma g(\mathbf{w}_t; \mathbf{z}_t)$$
$$\mathbf{v}_{t+1} = (1 - \beta_0)\mathbf{v}_t + \beta_0 \nabla g(\mathbf{w}_t; \mathbf{z}_t)\nabla f(s_{t+1})$$
$$\mathbf{w}_{t+1} = \mathbf{w}_t - \eta_0 \mathbf{v}_{t+1}$$

When $\eta_0\beta_0 = \eta$, $\beta = 1 - \beta_0$, the above two updates are equivalent. Hence, below we will analyze the second update.

Hence Lemma 1, 2 and 3 are applicable to ABSGD updates with $\tilde{\eta} = \eta_0, c_l = c_u = 1, \tilde{\beta} = \beta_0$. Based on this, we provide the convergence analysis for Theorem 1.

*Proof of Theorem 1.* Then by $\Delta_t = \|\nabla F(\mathbf{w}_t) - \mathbf{v}_{t+1}\|^2$ and according to Lemma 2, we have

$$\mathbb{E}[\Delta_{t-1}] \le \mathbb{E}\left[\frac{\Delta_{t-1} - \Delta_t}{\beta_0} + \frac{4L_F^2\eta_0^2\|\mathbf{v}_t\|^2}{\beta_0^2} + 4\beta_0 G^2 + 5G\|s_{t+1} - g(\mathbf{w}_t)\|^2\right]$$

Taking summation, we have

$$\mathbb{E}[\sum_{t=0}^{T-1}\Delta_t] \le \mathbb{E}\left[\sum_{t=0}^{T-1}\frac{\Delta_t - \Delta_{t+1}}{\beta_0} + \sum_{t=0}^{T-1}\frac{4L_F^2\eta_0^2\|\mathbf{v}_{t+1}\|^2}{\beta_0^2} + 4\beta_0 G^2 T + 5G\sum_{t=0}^{T-1}\|s_{t+2} - g(\mathbf{w}_{t+1})\|^2\right] \quad (13)$$

Next we bound $\mathbb{E}[\sum_{t=1}^{T}\|s_{t+1} - g(\mathbf{w}_t)\|^2] = \mathbb{E}[\sum_{t=0}^{T-1}\|s_{t+2} - g(\mathbf{w}_{t+1})\|^2]$ by the following Lemma 3:

$$\mathbb{E}_t[\|s_{t+1} - g(\mathbf{w}_t)\|^2] \le (1-\gamma)\|s_t - g(\mathbf{w}_{t-1})\|^2 + \gamma^2 V_g + \frac{2L_g^2\|\mathbf{w}_t - \mathbf{w}_{t-1}\|^2}{\gamma}.$$

$$\|s_t - g(\mathbf{w}_{t-1})\|^2 \le \frac{(\|s_t - g(\mathbf{w}_{t-1})\|^2 - \mathbb{E}_t[\|s_{t+1} - g(\mathbf{w}_t)\|^2])}{\gamma} + \gamma V_g + \frac{2L_g^2\|\mathbf{w}_t - \mathbf{w}_{t-1}\|^2}{\gamma^2}.$$

As a result,

$$\mathbb{E}_t[\sum_{t=1}^{T}\|s_{t+1} - g(\mathbf{w}_t)\|^2] \le \frac{\mathbb{E}[\|s_2 - g(\mathbf{w}_1)\|^2]}{\gamma} + \gamma V_g T + \sum_{t=1}^{T}\frac{2L_g^2\eta_0^2\|\mathbf{v}_t\|^2}{\gamma^2}. \quad (14)$$

Combining the equation (13) and (14) inequalities together we have

$$\mathbb{E}[\sum_{t=0}^{T-1}\Delta_t] \le \mathbb{E}\left[\frac{\Delta_0}{\beta_0} + \sum_{t=0}^{T-1}\frac{L_F^2\eta_0^2\|\mathbf{v}_{t+1}\|^2}{\beta_0^2} + 4\beta_0 G^2 T + \frac{5G\mathbb{E}[\|s_2 - g(\mathbf{w}_1)\|^2]}{\gamma} + \sum_{t=1}^{T}\frac{10GL_g^2\eta_0^2\|\mathbf{v}_t\|^2}{\gamma^2} + 5G\gamma V_g T\right] \quad (15)$$

Finally, combining Equation (15) with Lemma 1, and $\beta_0 = \gamma$, we have

$$\mathbb{E}\left[\frac{1}{T}\sum_{t=0}^{T-1}\|\nabla F(\mathbf{w}_t)\|^2\right] \le \frac{F(\mathbf{w}_1) - F(\mathbf{w}_T)}{\eta_0 T} + \frac{1}{T}\sum_{t=0}^{T-1}\frac{\Delta_t}{2} - \frac{1}{T}\sum_{i=0}^{T-1}\frac{\|\mathbf{v}_{t+1}\|^2}{4}$$

$$\le \frac{F(\mathbf{w}_1) - F_*}{\eta_0 T} + \frac{\mathbb{E}[\Delta_0 + 5\|s_2 - g(\mathbf{w}_1)\|^2]}{\gamma T} + \gamma(4G^2 + 5GV_g)$$

$$+ \mathbb{E}\left[\frac{1}{T}\sum_{t=0}^{T-1}\frac{(L_F^2 + 10GL_g^2)c_u^2\eta^2\|\mathbf{v}_{t+1}\|^2}{\gamma^2} - \frac{1}{4}\|\mathbf{v}_{t+1}\|^2\right].$$

where $F(\mathbf{w}_1) - F_* \le \Delta_F$, $\Delta_0 = \|\mathbf{v}_1 - \nabla F(\mathbf{w}_0)\|^2 = \|\beta_0\nabla g(\mathbf{w}_0; \mathbf{z}_0)\nabla f(s_1) - \nabla F(\mathbf{w}_0)\|^2 = \|\beta_0\nabla g(\mathbf{w}_0; \mathbf{z}_0)\nabla f(\gamma g(\mathbf{w}_0; \mathbf{z}_0)) - \nabla F(\mathbf{w}_0)\|^2 \le 2\beta_0^2 C_g^2 C_f^2 + 2C_F^2 \overset{\beta_0 \le 1}{\le} 2G^2 + 2C_F^2$, and $\|s_2 - g(\mathbf{w}_1)\|^2 =$

$\|(1-\gamma)s_1 + \gamma g(\mathbf{w}_1, \mathbf{z}_1) - g(\mathbf{w}_1)\|^2 \overset{s_0=0}{=} \|(1-\gamma)(\gamma g(\mathbf{w}_0, \mathbf{z}_0)) + \gamma g(\mathbf{w}_1, \mathbf{z}_1) - g(\mathbf{w}_1)\|^2 \overset{|a+b|^2 \leq 2a^2+2b^2}{\leq} 4C_0^2 + 4C_0^2 + 2C_0^2 = 10C_0^2.$

Then by setting $\beta_0 = \gamma \leq \frac{\epsilon^2}{3(4G^2+5GV_g)}, \eta_0 = \frac{\gamma}{2\sqrt{L_F^2+10GL_g^2}}, T \geq \max\{\frac{3(2G^2+2C_F^2+10C_0^2)}{\epsilon^2\gamma}, \frac{6\sqrt{L_F^2+10GL_g^2}\Delta_F}{\epsilon^2\gamma}\} = \max\{\frac{9(4G^2+5GV_g)(2G^2+2C_F^2+10C_0^2)}{\epsilon^4}, \frac{18(4G^2+5GV_g)\sqrt{L_F^2+10GL_g^2}\Delta_F}{\epsilon^4}\}, \eta_0 L_F \leq \frac{c_l}{2c_u^2}, \eta_0 L_F \leq \frac{1}{2}$, we have that

$$\mathbb{E}\left[\frac{1}{T}\sum_t \|\nabla F(\mathbf{w}_t)\|^2\right] \leq \epsilon^2,$$

Therefore, by $\eta = \eta_0\gamma, \beta = 1 - \gamma$, which finishes the proof of ABSGD in Theorem 1

$\square$

## 8.2 Convergence Analysis of ABAdam

**Theorem 2.** *Assume assumption 1 holds and there exists $C_0, C_1$ such that $\exp(L(\mathbf{w}_t; \mathbf{z}_i)/\lambda) \leq C_0, \|\nabla L(\mathbf{w}_t; \mathbf{z}_i)\| \leq C_1$, for all $\mathbf{w}_t$ and any $\mathbf{z}_i$, Then, For $\lambda = \tau > 0$ with appropriate $\eta, \gamma, \beta_1, \beta_2$, ABAdam ensures that $E\left[\frac{1}{T}\sum_{t=1}^{T}\|\nabla F_\tau^{(1)}(\mathbf{w}_t)\|^2\right] \leq \epsilon^2$ after $T = O(1/\epsilon^4)$ iterations, and for $\lambda = -\tau < 0$ with appropriate $\eta, \gamma, \beta$, ABAdam ensures that $\mathbb{E}\left[\frac{1}{T}\sum_{t=1}^{T}\|\nabla F_\tau^{(2)}(\mathbf{w}_t)\|^2\right] \leq \epsilon^2$ after $T = O(1/\epsilon^4)$ iterations, where we exhibit the constant in the big O.*

**ABAdam** The updates of ABAdam:

$$s_{t+1} = (1-\gamma)s_t + \gamma g(\mathbf{w}_t; \mathbf{z}_t)$$
$$\mathbf{v}_{t+1} = \beta_1 \mathbf{v}_t + (1-\beta_1)\nabla g(\mathbf{w}_t; \mathbf{z}_t)\nabla f(s_{t+1})$$
$$\mathbf{u}_{t+1} = \beta_2 \mathbf{u}_t + (1-\beta_2)(\nabla g(\mathbf{w}_t; \mathbf{z}_t)\nabla f(s_{t+1}))^2$$
$$\mathbf{w}_{t+1} = \mathbf{w}_t - \eta\left(\frac{\mathbf{v}_{t+1}}{\sqrt{\mathbf{u}_{t+1}} + G_0}\right)$$

To provide theoretical analysis for ABAdam, we add another assumption following the proof of (Guo et al., 2021).

**Proposition 3.** *Suppose Assumption 1 holds, there exists constant $c_l = 1/(G_0 + G), G = \max(C_g^2, C_f^2)$, and $c_u = \frac{1}{G_0}$, then $\mathbf{q}_t = 1/\sqrt{\mathbf{u}_t + G_0}$ is lower and upper bounded, such that $\forall i, c_l \leq \|\mathbf{q}_{t,i}\| \leq c_u$, where $\mathbf{q}_{t,i}$ denotes the i-th element of $\mathbf{q}_t$.*

*Proof.* Then

$$\|\nabla f(s_{t+1})\nabla g(\mathbf{w}_t; \mathbf{z}_t) \odot \nabla f(s_{t+1})\nabla g(\mathbf{w}_t; \mathbf{z}_t)\|_\infty \leq \|\nabla f(s_{t+1})\nabla g(\mathbf{w}_t; \mathbf{z}_t)\|^2 \leq C_g^2 C_f^2 \leq G^2$$

where $\odot$ represents Hadamard product. By set $u_{0,i} = \frac{C_0^2}{\lambda^2}G_\infty^2$, $\mathbf{u}_{t+1,i} = (1 - \beta_2)^t u_{0,i} + \beta_2\|(\nabla f(s_{t+1})\nabla g(\mathbf{w}_t; \mathbf{z}_t))^2\|_\infty \leq G^2 \ \forall t \geq 0$. Therefore $\frac{1}{G_0} \geq \frac{1}{\sqrt{\mathbf{u}_{t+1,i}+G_0}} \geq \frac{1}{G_0+G}$. $\square$

*Proof of ABAdam.* The proof of ABAdam can reuse the steps of ABSGD up to Equation (15) by setting $\beta_1 = \beta_0, \eta_0 = \tilde{\eta}$. In addition, by setting $\eta c_l \leq \tilde{\eta} = \frac{\eta}{G_0+\sqrt{\mathbf{u}_{t,i}}} \leq \eta c_u$

Then according to proposition 3 and Lemma 1, and let $\gamma = 1 - \beta_1$, we have

$$\mathbb{E}\left[\frac{1}{T}\sum_{t=0}^{T-1}\|\nabla F(\mathbf{w}_t)\|^2\right] \leq \frac{F(\mathbf{w}_1) - F(\mathbf{w}_T)}{c_l \eta T} + \frac{c_u}{c_l T}\sum_{t=0}^{T-1}\frac{\Delta_t}{2} - \frac{1}{T}\sum_{i=0}^{T-1}\frac{\|\mathbf{v}_{t+1}\|^2}{4}$$

$$\leq \frac{F(\mathbf{w}_1) - F_*}{c_l \eta T} + \frac{c_u \mathbb{E}[\Delta_0 + 5\|s_2 - g(\mathbf{w}_1)\|^2]}{\gamma T c_l} + \gamma(4G^2 + 5GV_g)\frac{c_u}{c_l}$$

$$+ \mathbb{E}\left[\frac{1}{T}\sum_{t=0}^{T-1}\frac{c_u^3(L_F^2 + 10GL_g^2)\eta^2\|\mathbf{v}_{t+1}\|^2}{c_l \gamma^2} - \frac{1}{4}\|\mathbf{v}_{t+1}\|^2\right].$$

The same as ABSGD, $F(\mathbf{w}_1) - F_* \leq \Delta_F$, $\Delta_0 \leq 2G^2 + 2C_F^2$, $\|s_2 - g(\mathbf{w}_1)\|^2 \leq 10C_0^2$. Then by setting $1 - \beta_1 = \gamma \leq \frac{\epsilon^2 c_l}{3c_u(4G^2 + 5GV_g)}, \eta = \frac{\sqrt{c_l}\gamma}{2c_u^{3/2}\sqrt{L_F^2 + 10GL_g^2}}, T \geq \max\{\frac{3c_u(2G^2 + 2C_F^2 + 10C_0^2)}{\epsilon^2 c_l \gamma}, \frac{6\sqrt{L_F^2 + 10GL_g^2}\Delta_F}{\epsilon^2 c_l \gamma}\} = \max\{\frac{9c_u^2(4G^2 + 5GV_g)(2G^2 + 2C_F^2 + 10C_0^2)}{\epsilon^4 c_l^2}, \frac{18c_u(4G^2 + 5GV_g)\sqrt{L_F^2 + 10GL_g^2}\Delta_F}{\epsilon^4 c_l^2}\}, \eta_0 L_F \leq \frac{c_l}{2c_u^2}$, we have that

$$\mathbb{E}\left[\frac{1}{T}\sum_t\|\nabla F(\mathbf{w}_t)\|^2\right] \leq \epsilon^2,$$

which finishes the proof of ABAdam in Theorem 1. $\qquad\square$

## 8.3 Equivalence derivation between Min-max Robust Optimization and Composition formulation

In the section, we show the equivalence between the min-max formulation (3), (4), i.e,

$$\min_{\mathbf{w}\in\mathcal{R}^d}\max_{\mathbf{p}\in\Delta_n}\underbrace{\sum_{i=1}^n p_i L(\mathbf{w};\mathbf{z}_i) - \tau\sum_i p_i \ln(np_i)}_{F_\tau^{(1)}(\mathbf{w})} + r(\mathbf{w}), \tau > 0$$

$$\min_{\mathbf{w}\in\mathcal{R}^d}\min_{\mathbf{p}\in\Delta_n}\underbrace{\sum_{i=1}^n p_i L(\mathbf{w};\mathbf{z}_i) + \tau\sum_i p_i \ln(np_i)}_{F_\tau^{(2)}(\mathbf{w})} + r(\mathbf{w}), \tau < 0$$

and the composition formulations, i.e,

$$F_\tau^{(1)}(\mathbf{w}) = \tau\log\frac{1}{n}\sum_i\exp(L(\mathbf{w};\mathbf{z}_i)/\tau) + r(\mathbf{w}), \quad \tau > 0$$

$$F_\tau^{(2)}(\mathbf{w}) = -\tau\log\frac{1}{n}\sum_i\exp(-L(\mathbf{w};\mathbf{z}_i)/\tau) + r(\mathbf{w}), \quad \tau < 0$$

*Proof.* Here, we provide the detailed derivation for $\tau > 0$. Similarly, the derivation for $\tau < 0$ can be done using the same method. Recall the problem:

$$\min_{\mathbf{w}\in\mathcal{R}^d}\max_{\mathbf{p}\in\Delta_n} F_{\mathbf{p}}(\mathbf{w}) = \sum_{i=1}^n p_i\ell(\mathbf{w};\mathbf{z}_i) - h(\mathbf{p},\mathbf{1}/n) + r(\mathbf{w}),$$

where $\Delta_n = \{\mathbf{p}\in\mathcal{R}^n : \sum_i p_i = 1, 0 \leq p_i \leq 1\}$. In order to solve the inner maximization, we will fix $\mathbf{w}$ and derive an optimal solution $\mathbf{p}^*(\mathbf{w})$ that depends on $\mathbf{w}$. To this end, we consider the following problem:

$$\min_{\mathbf{p}\in\Delta_n} -\sum_{i=1}^n p_i\ell(\mathbf{w};\mathbf{z}_i) + h(\mathbf{p},\mathbf{1}/n)$$

where $r(\mathbf{w})$ was neglected since it does not involve $\mathbf{p}$. Note the expression of $h(\mathbf{p},\mathbf{1}/n) = \tau\sum_i p_i\log(np_i) = \tau\sum_i p_i\log(p_i) + \tau\log(n)$ due to $\sum_i p_i = 1$. There are three constraints to handle, i.e., $p_i \geq 0, \forall i$ and

$p_i \leq 1, \forall i$ and $\sum_i p_i = 1$. Note that the constraint $p_i \geq 0$ is enforced by the term $p_i \log(p_i)$, otherwise the above objective will become infinity. As a result, the constraint $p_i < 1$ is automatically satisfied due to $\sum_i p_i = 1$ and $p_i \geq 0$. Hence, we only need to explicitly tackle the constraint $\sum_i p_i = 1$. To this end, we define the following Lagrangian function

$$L_{\mathbf{w}}(\mathbf{p}, \mu) = -\sum_{i=1}^{n} p_i \ell(\mathbf{w}; \mathbf{z}_i) + \tau(\log n + \sum_i p_i \log(p_i)) + \mu(\sum_i p_i - 1)$$

where $\mu$ is the Lagrangian multiplier for the constraint $\sum_i p_i = 1$. The optimal solutions satisfy the KKT conditions:

$$-\ell(\mathbf{w}; \mathbf{z}_i) + \tau(\log(p_i^*(\mathbf{w})) + 1) + \mu = 0,$$
$$\sum_i p_i^*(\mathbf{w}) = 1$$

From the first equation, we can derive $p_i^*(\mathbf{w}) \propto \exp(\ell(\mathbf{w}; \mathbf{z}_i)/\tau)$. Due to the second equation, we can conclude that $p_i^*(\mathbf{w}) = \frac{\exp(\ell(\mathbf{w}; \mathbf{z}_i)/\lambda)}{\sum_i \exp(\ell(\mathbf{w}; \mathbf{z}_i)/\lambda)}$. Plugging this optimal $\mathbf{p}^*(\mathbf{w})$ into the original min-max objective, we have

$$\sum_{i=1}^{n} p_i^*(\mathbf{w}) \ell(\mathbf{w}; \mathbf{z}_i) - \tau(\log n + \sum_i p_i^*(\mathbf{w}) \log(p_i^*(\mathbf{w}))) + r(\mathbf{w}) = \tau \log \frac{1}{n} \sum_i \exp(\ell(\mathbf{w}; \mathbf{z}_i)/\tau) + r(\mathbf{w}),$$

which is the $F_\tau^{(1)}(\mathbf{w})$. $\qquad \square$

