# OpenReview forum: "Attentional-Biased Stochastic Gradient Descent"
_TMLR — Accepted by TMLR_

### Review · Reviewer_Nn3h · 2023-01-24

**Summary Of Contributions:**

The paper introduces an example weighting approach to learning from class-imbalanced data (or hard examples) as well as noisy examples. The method consists of assigning weights to the examples in a batch based on their loss value. The form of the weights mimics the exponentiated gradient update. However, the denominator is replaced by an EMA instead of the sum of the numerator. A convergence analysis of the algorithm is provided, and the results are validated on standard datasets.

**Audience:**

Yes

**Claims And Evidence:**

No

**Requested Changes:**

- Please provide a better motivation for using an EMA instead of the sum for normalizing the weights. The current motivation based on alternating between minimizing the loss over all examples vs. the examples in the batch is inconsistent. If the EMA does not have a proper theoretical motivation but shows significant improvements in practice, please show this empirically and perform a thorough comparison with CIW.
- Please verify the experimental results using EG. The results seem to be inconsistent with the ones reported by Kumar and Amid (2021).
- Please draw a distinction between the results in Theorem 1 and the results in Bar et al. (2021), Theorem 4.
- Please use the standard (author, year) bibliography format of TMLR.
- Please replaces $\text{ln}$ with $\log$.
- Please add the robust bi-tempered loss (Amid et al., 2019) and more examples of robust losses to the references (see Kumar and Amid (2021)).

**Strengths And Weaknesses:**

### Major Comments
- The proposed method is strikingly similar to CIW (Kumar and Amid, 2021), where the updates are motivated via a min-min formulation with an $\alpha$-divergence constraint on the weights (although the authors seem to have missed this work). $\alpha=1$ corresponds to the KL divergence and the Exponentiated Gradient (EG) updates in the current paper. However, the only difference between the proposed method is the denominator. In CIW, the denominator is set to the sum of the numerator terms, therefore, yielding a probability distribution for the weights. In the proposed approach, the denominator is replaced with an EMA over training batches. However, my main concern is with the correctness of this replacement. In Eq. (6), the minimization is given over all examples in the training set, thus, motivating Eq. (8), where the denominator is replaced with $s_{t+1}$. However, the actual update is given in Eq. (2), which is over the mini-batch of examples $B$. Thus, I find this motivation conflicting, and I don't see any reason to use a different normalizer than the one in CIW. If the authors believe that this heuristic change works significantly better in practice, they should perform a clear comparison with CIW with and without this alteration.
- The min-max formulation can also be derived from CIW, although no experiments are given for the min-max formulation in the CIW paper.
- The convergence results for Majidi et al. (2021) are given by Bar et al. (2021) (Theorem 4), which is essentially the same result in the current work.
- The additional derivation for ABAdam in Eq. (9) is essentially redundant. Basically, the reweighted gradient can be passed into any optimizer (including Adam) without any alteration to the optimizer. The update in Eq. (9) precisely yields this procedure.


### Minor Comments
- Please use the standard (author, year) bibliography format of TMLR.
- Please use $\log$ instead of $\text{ln}$ consistently throughout (Eq. (6) and Eq. (7)).
- The robust bi-tempered loss (Amid et al., 2019) is the correct generalization of the cross entropy loss (in terms of properness). Please also include it in the related work on robust losses.
- The experimental results using EG seem to be weaker than expected. The results are in contrast with the ones reported by Kumar and Amid (2021). EG has a long-term memory (at the cost of storing one weight per example) and, thus, works better than the reweighting methods that rely on only batch information. The EMA in the proposed method is unlikely to produce such improvement (since it aggregates information over different batches, not the same examples in the batch over multiple trials). Please check your tuning procedure and make sure the learning rate schedules for the EG weights are consistent with Majidi et al. (2021).

### References
- Abhishek Kumar and Ehsan Amid. "Constrained Instance and Class Reweighting for Robust Learning under Label Noise." arXiv preprint arXiv:2111.05428 (2021).
- Noga Bar, Tomer Koren, and Raja Giryes. "Multiplicative Reweighting for Robust Neural Network Optimization." arXiv preprint arXiv:2102.12192 (2021).
- Ehsan Amid, Manfred K. Warmuth, Rohan Anil, and Tomer Koren. "Robust Bi-Tempered Logistic Loss Based on Bregman Divergences." Neurips (2019).

---

> ### Author Response · Authors · 2023-03-21
> **Thanks for the suggestion and comments!**
>
> > **Q1**:  The first major concern:  The proposed method is strikingly similar to CIW (Kumar and Amid, 2021), where the updates are motivated via a min-min formulation with an  $\alpha$-divergence constraint on the weights.  $\alpha=1$ corresponds to the KL divergence and the Exponentiated Gradient (EG) updates in the current paper. ... In CIW...."
>
> > **Answer**: Thank you for pointing out the CIW paper. **First**, we would like to make several clarifications: (i) our work is a parallel work of CIW (our first version appears online in Dec. 2020 and the CIW paper appears online in Nov. 2021 though our first version does not consider the min-min formulation); (ii) the normalization by using $s_{t+1}$ is not a heuristic. This is motivated by solving the compositional objective as in equation (6) in our paper. We have a rigorous convergence result of our method using $s_{t+1}$ for normalization in Theorem 1 (its proof was originally included in the supplementary material, and is included in the appendix after reference in the revision). We refer the reviewer to our convergence analysis before considering it as a heuristic.  In contrast, the CIW paper does not discuss any convergence results. The naive approach of simply using the sum of numerator terms of all data is expensive as it requires calculating the losses of all data at each iteration. The authors of CIW have suggested using the mini-batch data for constructing the loss. However, there is no convergence guarantee of this approach that uses the data in the mini-batch for constructing the loss. A similar approach to CIW was used in Majidi et al. 2021. In our paper, we have compared with  Majidi et al. 2021. Please refer to Table 7 for the comparison where EG is the method of Majidi et al. In the revision, we have implemented CIW with $\alpha$-divergence ($\alpha=1$) and included its result in Table 7 for Clothing1M data.   **Second**, in the revision we compared our normalization with the naive mini-batch normalization, where the latter corresponds to setting $\gamma=1$ in our method. In Figure 4 in our revision, we have compared our method with different values of $\gamma$ varying from 0.1 to 1, which clearly shows that using $s_{t+1}$ with $\gamma<1$ for normalization is better than using the mini-batch normalization (corresponding to $\gamma=1$).  **Third**, the mini-batch version in Eq (2) is not conflicting with our motivation in Eq (8). If using one example at each iteration, we can use $\widetilde p_i \nabla L(\mathbf w_t; \mathbf z_i)$ as the gradient estimator.  In practice, we usually use a mini-batch and hence we take the average of the mini-batch data for constructing the gradient estimator. Eq (2) is using the mini-batch gradient estimator  $\frac{1}{B}\sum_{i=1}^B\widetilde p_i \nabla L(\mathbf w_t; \mathbf z_i)$ and at the same time using a momentum update. Please refer to the paragraph above Sec. 3.4 for the motivation of using the momentum update. The experimental results in Figure 5 in the appendix demonstrate the benefit of using the momentum update compared with no momentum update.
>
> >**Q2**: The convergence results for Majidi et al. (2021) are given by Bar et al. (2021) (Theorem 4), which is essentially the same result in the current work.
>
> >**Answer**: Thank you for pointing out the work of Bar et al (2021).   We do not agree that the convergence of Majidi et al's EG method is proved by Bar et al. (2021) (Theorem 4). In the latter work, their algorithm is different from the EG method in  Majidi et al. In particular,  Algorithm 1 in Bar et al. needs to project the weights for all examples into a constrained  simplex which takes O($n^2$) complexity as mentioned in their paper, where $n$ is the size of the training set. However, the EG method in Majidi et al.  does not require such projection.  In addition, in Bar et al. (2021) (Theorem 4), $T$ represents $T$ epochs with a total of $\mathcal T=Tn/B$ iterations  in their Algorithm 1, where $B$ is the batch size.  Hence, the convergence rate Theorem 4 in terms of iterations is $O(\sqrt{1/T})=O(\sqrt{\frac{n}{B\mathcal T}})$, which is actually  $n/B$ times worse than the rate of our algorithm in Theorem 1.
>
>
> > **Q3**: The robust bi-tempered loss (Amid et al., 2019) is the correct generalization of the cross entropy loss (in terms of properness). Please also include it in the related work on robust losses.
>
> >**Answer**: We added the reference to the related work
>
>
> > **Q4**: The experimental results using EG seem to be weaker than expected. The results are ... EG weights are consistent with Majidi et al. (2021).
>
> >**Answer**: Thanks for the suggestion!
> To be consistent with Majidi et al. (2021), we adjust the network from ResNet34 to ResNet18 for the asymmetric noisy setting of CIFAR100. Then by finetuning the learning rates and weight decay parameters, we report the final results in **Table 6** the revision. In addition, we add the baseline of CIW with $\alpha = 1$.

---

### Review · Reviewer_HCLQ · 2023-01-25

**Summary Of Contributions:**

This work proposes an end-to-end SGD variant namely ABSGD to handle the problem of data imbalance and label noises. With comprehensive theoretical analysis on the model update and experiments on multiple benchmark datasets, the paper demonstrates that ABSGD shows better performance than baselines in terms of the effectiveness.

**Audience:**

Yes

**Claims And Evidence:**

Yes

**Requested Changes:**

Please address the weak points aforementioned in the revision.

**Strengths And Weaknesses:**

Strong points:

(1) the motivation of this paper is clear and meaningful, where the label imbalance and noisy problems in DL are critical and challengeable especially considering the efficiency as well.

(2) the technical part is sound and without obvious flaw for me.


Weak points:

(1) there are several typos, grammar issues, and layout problems (e.g., Table 8 is totally missing) through the manuscript.

(2) since the meta-learning based method is the main competitive baseline (from the prospect of accuracy compared with ABSGD), it is recommended to adopt other mainstream backbones (e.g., transformer-based architectures) instead of only comparing the performance with ResNet series to demonstrate the robustness of ABSGD.

(3) As mentioned in this paper, compared with existing individual-level weighting methods using meta-learning that require three backward propagations for computing mini-batch stochastic gradients, ABSGD is more efficient with only one backward propagation at each iteration as in standard deep learning methods. In other words, the efficiency improvement could be reflected in the training runtime, where there is no related experiments justified.

---

> ### Author Response · Authors · 2023-03-21
> **Thanks for the suggestion and comments!**
>
>  > **Q1**:there are several typos, grammar issues, and layout problems (e.g., Table 8 is totally missing) through the manuscript.
>
> >**Answer**: We have gone through the paper again and corrected some typos and grammar issues. Table 8 was originally included in the Appendix, which is in the supplementary file. In the revision, we have included the Appendix after the reference.
>
> >**Q2**: since the meta-learning-based method is the main competitive baseline (from the prospect of accuracy compared with ABSGD), it is recommended to adopt other mainstream backbones (e.g., transformer-based architectures) instead of only comparing the performance with the ResNet series to demonstrate the robustness of ABSGD.
>
> >**Answer**: Thanks for the suggestion! Due to the memory concern, we cannot use mainstream transformer-based architectures, e.g., ViT-B/16 has 86M parameters, which cannot be operated in our NVIDIA GeForce GTX 1080 Ti with 11 GB memory. Instead, we have implemented the newly proposed structure, named as Convmixer~[r1], which operates convolutional layers on small patches. Convmixer has been shown to achieve competitive results as ViT models but with faster training speed and fewer parameters. We conducted an experiment by comparing  ABSGD with CE loss and SGD for optimizing CE loss with on CIFAR10 dataset in the long tail setting with an imbalance ratio of 10, and 100. The result is presented in the following table.  We have include this result in Table 10 in the appendix of our revision.
>         &
> | Imbalance Ratio | SGD (CE) | ABSGD (CE)|
> | --- | ----------- |-----|
> | 10  | 82.1  | 83.9|
> | 100 | 63.2  | 66.31|
>
> [r1]: \href{Patches Are All You Need? }\url{https://arxiv.org/pdf/2201.09792.pdf}
>
> >**Q3**: As mentioned in this paper, compared with existing individual-level weighting methods using meta-learning that require three backward propagations for computing mini-batch stochastic gradients, ABSGD is more efficient with only one backward propagation at each iteration as in standard deep learning methods. In other words, the efficiency improvement could be reflected in the training runtime, where there is no related experiments justified.
>
> >**Answer**: We have conducted an experiment on CIFAR-10 data with different networks on NVIDIA GeForce GTX 1080 Ti. The running time (seconds) per iteration of SGD, ABSGD and META methods are shown in the following table.  It is clear to see that ABSGD has a comparable running time as SGD, while the per iteration running time of META is more than 10 times larger than SGD and ABSGD. We have included this result in Table 9 in the appendix of our revision.
>
> | Network(\# Param.) | SGD| ABSGD|META|
> | --- | ----------- |-----|-----|
> | ResNet32 (0.46M)   | 0.0167       | 0.0176       | 0.376 |
> | ResNet44 (0.44M)  | 0.0234       | 0.0250         | 0.474 |
> | ResNet56 (0.85M)  | 0.0284       | 0.0296        | 0.566|
> | ResNet110 (1.7M)  |  0.0684       | 0.0692         | 0.882 |

---

> > ### Comment · Reviewer_HCLQ · 2023-03-28
> > **Authors' responses have addressed my concerns.**
> >
> > Authors' responses havea already addressed my concerns.

---

### Review · Reviewer_c8Ys · 2023-02-20

**Summary Of Contributions:**

This paper proposed a modification of gradient-based optimization methods, which is claimed to be effective for label imbalance and label noise. The proposed method essentially combines a loss function modification and an adaptive learning rate. The proposed method was evaluated on the image classification task.

**Audience:**

Yes

**Broader Impact Concerns:**

The author did not provide a broader impact statement.

**Claims And Evidence:**

No

**Requested Changes:**

The explanation of why the proposed method can solve label imbalance/noise problems and the minor issues listed above.

**Strengths And Weaknesses:**

## Strengths

- The proposed method is simple and easy to implement.
- The modification can be applied to multiple gradient-based optimization methods.

## Weaknesses

- It is nice that the author aimed to solve two problems: label imbalance and label noise. However, it is weird that the author treated them as two disjoint problems (Eqs. (3) and (4), depending on positive or negative $\lambda$). In some real-world situations, both can happen at the same time. Based on the alleged dichotomy, it is unclear how to deal with the case when imbalance and noise are present.
- Many existing works on noisy labels assume a noise model (symmetric, class-conditional, instance-dependent, etc.), but it seems this work did not specify one. Hence, it is unclear what kind of noisy label problems can be solved by the proposed method.

## Technical issues

The presentation of the proposed method is a bit confusing. The weight regularization and the momentum term seem not essential to the proposed method. The weight $\tilde{p}_i$ in Eq. (2), which is the novel part of the proposed method, is a fraction of an $i$-dependent part $\exp(\frac{L\_i}{\lambda})$ and an $i$-independent part $s\_{t+1}$. If I did not misunderstand anything, the update step contributed by the $i$-th example is
\begin{equation}
\frac{\eta}{B s\_{t+1}} e^\frac{L\_i}{\lambda} \nabla L\_i = \frac{\eta}{B s\_{t+1}} \nabla (\lambda e^\frac{L\_i}{\lambda})
\end{equation}
Then, we can see that the proposed method essentially changes the loss function from $L$ to $\lambda e^\frac{L}{\lambda}$ and uses an adaptive learning rate based on the reciprocal of a moving average of the empirical risk of each batch.

Then, why does the proposed method solve label imbalance or label noise problems? The theoretical analysis provides little insight.

## Minor issues

- The term "attentional-biased" is confusing because it hints at the attention architecture (e.g., transformer), but it is only reweighting here.
- The author claimed that the method is "systematic," but its meaning is unclear.
- The author used Apple's FaceID and autonomous driving cars as examples in the introduction section, but they seem not to be classification problems studied in this work.
- "Existing studies in these methods are not very successful for deep learning with big data": insufficient support.
- As far as I know, "robust weighting" is not widely accepted in the community and was not properly defined in this paper.
- "The focal loss lacks theoretical foundation": see https://arxiv.org/abs/2011.09172.
- "Most existing optimization algorithms for DRO are not practical for deep learning": insufficient support.
- A [simplex](https://en.wikipedia.org/wiki/Simplex) in $\mathbb{R}^n$ is $(n-1)$-dimentional.
- Step 7?
- $\lambda \in \\{\mathbb{R} \setminus 0\\}$ should be $\lambda \in \mathbb{R} \setminus \\{0\\}$
- Do $F_\tau^{(1)}(\mathbf{w})$ and $F_\tau^{(2)}(\mathbf{w})$ include $r(\mathbf{w})$ or not?
- The expression $\min A = B$ is ambiguous, which means $B$ is the minimal value of $A$, but it seems that $B$ is defined as or equal to $A$ here.

---

> ### Author Response · Authors · 2023-03-21
> **Thanks for the comments and suggestions!**
>
> > **Q1**: It is nice that the author aimed to solve two problems: label imbalance and label noise. However, it is weird that the author treated them as two disjoint problems (Eqs. (3) and (4), depending on positive or negative. In some real-world situations, both can happen at the same time. Based on the alleged dichotomy, it is unclear how to deal with the case when imbalance and noise are present.
>
> >**Answer**:  We agree with the reviewer that the proposed method is not for tackling both data imbalance and noisy labels.  One way to tackle both issues is to use ABSGD with a positive $\lambda$ and a robust individual loss that is robust to label noise, e.g., the symmetric CE loss and the generalized CE loss [50, 51], i.e, Wang et al. (2019) and Zhang \& Sabuncu (2018).  We leave this as a future work.
>
> >**Q2**: Many existing works on noisy labels assume a noise model (symmetric, class-conditional, instance-dependent, etc.), but it seems this work did not specify one. Hence, it is unclear what kind of noisy label problems can be solved by the proposed method.
>
> >**Answer**: Existing works need to assume a particular noise model for crafting a specific robust loss function so that it can be proved to be robust under the considered noise setting. In contrast, our work is loss independent and can be leveraged for boosting existing works of robust losses. In our experiments, we compared ABSGD and SGD with different loss functions, including the symmetric CE loss [50], i.e, Wang et al. (2019), and the generalized CE loss [51], i.e, Zhang \& Sabuncu (2018),  which are proposed with the symmetry property in mind to be robust against symmetric and class-conditional noise. We considered both symmetric and class-conditional noise as in Table 6 and real noise setting as in Table 7.
>
> >**Q3**:
> The presentation of the proposed method is a bit confusing. The weight regularization and the momentum term seem not essential to the proposed method.
>
> >**Answer**: The weight regularization is included for more generality. The momentum term is essential to our method for proving our convergence. Please check the paragraph above sec. 3.4. In particular, if the momentum term is removed the convergence for solving Eq (6) is only $O(1/\epsilon^5)$ instead of $O(1/\epsilon^4)$ as we proved in the paper.
>
>
> >**Q4**: Then, we can see that the proposed method essentially changes the loss function from $L$
>  to $\lambda e^{L/\lambda}$ and uses an adaptive learning rate based on the reciprocal of a moving average of the empirical risk of each batch. Then, why does the proposed method solve label imbalance or label noise problems? The theoretical analysis provides little insight.
>
>
> >**Answer**: We would like to draw the reviewer's attention that our algorithm is tied to solving the min-max or min-min DRO formulations (3) or (4). The theoretical result in Theorem 1 provides the convergence guarantee of optimization. The motivation of using min-max or min-min DRO for tackling data imbalance and label noises have been demonstrated in previous works, which is not in our scope. For example, Shalev-Shwartz & Wexler (’16 ICML) has shown that the min-max DRO formulation ($\lambda$ approaches 0) is good for learning with imbalanced data. The robustness to outliers of titling (equivalent to (6)) has been studied in many papers, dating back to earlier 2000s (e.g., [65], i.e, Choi et al. (2000)), and modern studies have been considered in [59], i.e,  Li et al. (2021).
>
>
> > **Regarding the Minor issues.**
> - The term "attentional-biased" is confusing because it hints at the attention architecture (e.g., transformer), but it is only reweighting here.
>   * **Answer**: We note that the word \"attentional\" is not typically used in existing works of attention-based networks which typically use  the noun ``attention" instead of adjective attentional. We use the adjective \"attentional-biased\" to better reveal the feature of our method because the weight of each data in attentionally constructed such that the gradient estimator is not unbiased of the empirical risk.  Hence, we reserve the wording \"attentional\" in the revision.
> - The author claimed that the method is "systematic," but its meaning is unclear.
>   *  **Answer**:  We change the \"systematic\" to \"provable\".
> - The author used Apple's FaceID and autonomous driving cars as examples in the introduction section, but they seem not to be classification problems studied in this work. Existing studies in these methods are not very successful for deep learning with big data": insufficient support
>      *  **Answer**:  These examples are used for layman such that they can quickly get a feeling of the problem that we want to tackle.

---

### Review · Reviewer_Mbkp · 2023-02-21

**Summary Of Contributions:**

This work proposes a variant of SGD that assigns different weights to different data points. The method is shown to be relevant for problems such as class imbalanced learning and label noise learning

**Audience:**

Yes

**Claims And Evidence:**

No

**Requested Changes:**

The authors might want to significantly improve the motivation part of the paper and align the results with the motivation. Currently, I do not find the motivation convincing, nor do the results answer the stated motivation

Theory-wise, it would be more convincing if the authors presented a theory/proof of how the proposed method improves the robustness of SGD in toy examples. Proof of convergence is nice, but does not help with the main claims of the paper

**Strengths And Weaknesses:**

Strength:
1. the paper accompanies the proposal with sound theory

Weakness:
1. lack of novelty. Ref. (59) and (66) already propose this method. In comparison, the authors argue that their proposal suggested how to incorporate momentum, but adding the momentum seems rather trivial.

2. motivation is poor. If the idea is to improve the robustness of SGD to data imbalance or label noise, the authors should first present some convincing evidence showing that SGD does have a serious problem here -- not just on a toy example, and the comparison is only made for the first iteration

3. in fact, the empirical result of this work implies one of the following two things: (1) SGD does not have a problem with data imbalance, or (2) the proposed method does not do better than SGD in dealing with SGD. See Table 1; even at an imbalance ratio of 100, the proposed method only improves SGD by 0.7%, which is actually with the error bar of the result for SGD. I see no significant difference between the proposed method and SGD

4. a question about the theory. in theorem 1, does one have to decrease the learning rate as a power to ensure convergence? If so, it should be stated clearly. Or does the theory apply to a constant learning rate?

---

> ### Author Response · Authors · 2023-03-21
> **Thanks for the comments and suggestions!**
>
> > **Q1**: lack of novelty. Ref. (59), i.e, Li et al. (2021) and (66), Majidi et al. (2021), already propose this method. In comparison, the authors argue that their proposal suggested how to incorporate momentum, but adding the momentum seems rather trivial.
>
> >**Answer**:  We would like to point out that our work is a concurrent work with [59].  The first version of our work was released  in December 2020, which is 10 months earlier than [59] which appears online in Oct. 2021. We did notice that they have an earlier conference version appearing online in July 2020, which however did not include the convergence result of their stochastic algorithm.  The main novelty of this paper is to demonstrate a simple algorithm (ABSGD) optimizes theoretically grounded DRO formulations with promising performance for imbalanced data and noisy labels, and to establish its convergence.  Our contributions compared with [59,66] include: (1) our method is simpler than that in [59]. As mentioned in the remark under Theorem 1, [59] only proves the convergence for the algorithm with independent mini-batches for computing $L(\mathbf w; \mathbf z)$ and $\nabla L(\mathbf w; \mathbf z')$. Instead, our algorithm uses the same mini-batch for calculating the losses and their gradients; (2) our work provides convergence guarantee for solving the min-min formulation, which was not provided in [66].
>
> >**Q2**: motivation is poor. If the idea is to improve the robustness of SGD to data imbalance or label noise, the authors should first present some convincing evidence showing that SGD does have a serious problem here -- not just on a toy example, and the comparison is only made for the first iteration
>
> >**Answer**: We add more references for the lack of robustness of SGD to data imbalance and label noise in the revision. In addition, on the toy example, we have provided the comparison of SGD and ABSGD for their final models in Appendix, which shows similar trends to Figure 1.
>
>
> >**Q3**: in fact, the empirical result of this work implies one of the following two things: (1) SGD does not have a problem with data imbalance, or (2) the proposed method does not do better than SGD in dealing with SGD. See Table 1; even at an imbalance ratio of 100, the proposed method only improves SGD by 0.7\%, which is actually with the error bar of the result for SGD. I see no significant difference between the proposed method and SGD.
>
> >**Answer**: Please note that we should look at the results on different datasets in different settings not just one dataset in one setting. In Table 1, on CIFAR-10 for imbalanced setting ST with imbalance ratio 100, our method improves SGD by 2-3\%. In Table 2 with label dependent losses, our method improves SGD by a large margin, e.g., even with more than 10\% improvement.   Table 3 shows our method has 7\% improvement compared with  the CBCE optimized by SGD. Table 4 and Table 5 also demonstrate significant improvements of our method.
>
>
>
> > **Q4**: a question about the theory. in theorem 1, does one have to decrease the learning rate as a power to ensure convergence? If so, it should be stated clearly. Or does the theory apply to a constant learning rate?
>
> >**Answer**: We have modified our theorem to reflect the setting of the learning rate and other hyper-parameters. In our analysis, we use a learning rate in the order of $O(\epsilon^2)$.
>
> >**Q5**:The authors might want to significantly improve the motivation part of the paper and align the results with the motivation. Currently, I do not find the motivation convincing, nor do the results answer the stated motivation.  Theory-wise, it would be more convincing if the authors presented a theory/proof of how the proposed method improves the robustness of SGD in toy examples. Proof of convergence is nice, but does not help with the main claims of the paper.
>
> >**Answer**: Please refer to response of Q2.

---

### Decision · Action_Editors · 2023-04-06

**Recommendation:** Accept with minor revision

**Comment:**

This paper designs an instance-wise weighting mechanism for stochastic optimization in gradient based algorithms for robust optimization. The algorithm can be explained by the min-max or min-min distributionally robust optimization principles, and is well justified by both theory and experiment results.

There are 4 reviewers, and they hold different opinions about the paper. I went through the paper, the comments and the rebuttal, and I believe most of the reviewer concerns have been addressed, and thus the paper is acceptable after a minor revision. Specifically, I summarize the main concerns and my opinions below:

Reviewer Mbkp: 1) motivation and novelty: the revision has incorporated more examples and detailed explanations, 2) experiments: more explainations and results are added in the revision. I believe the rebuttal has cleared out the problems.

Reviewer c8Ys: 1) separation of label imbalance and label noise: this is minor and can be resolved by more explanations; 2) technical issues: I believe the rebuttal has addressed the issues.

Reviewer HCLQ: more comparisons with existing methods: the rebuttal has added more experimental results, which I believe has addressed the problem.

Reviewer Nn3h: the reviewer has more fundamental problems, and he has communicated with the authors in the rebuttal. The interaction cleared most of the questions, but the reviewer still concerns about the following: 1) the calculating the normalization in CIW and EG (baselines) being expensive: I believe this is a misunderstanding, the authors did not claim this; 2) the proposed algorithm does not have normalized weights in a minibatch: I agree with the authors that this is unnecessary, and the reasoning of the authors is convincing; 3) why using the EMA for computing the normalization: the authors provide detailed explanations as well as theoretical guarantees, which are lack of in other methods, which I agree with; 4) methods of Majidi et al. and Bar et al. are the same: I agree with the authors there are some differences between these two algorithms, and in addition, this does not impact the contribution of the proposed method too much.

Overall, I believe the concerns raised by the reviewers have been well addressed in the rebuttal. However, I request the authors to incorporate the comments from the reviewers and revise their paper accordingly, to make it more complete and clear.


**Audience:**

Yes, researchers from the optimization community could be interested in the findings.

**Claims And Evidence:**

In my opinion, the claims made in the revision of this paper are supported by accurate and convincing evidence.